# Three-Dimensional Structural Heteromorphs of Mating-Type Proteins in *Hirsutella sinensis* and the Natural *Cordyceps sinensis* Insect–Fungal Complex

**DOI:** 10.3390/jof11040244

**Published:** 2025-03-23

**Authors:** Xiu-Zhang Li, Yu-Ling Li, Jia-Shi Zhu

**Affiliations:** State Key Laboratory of Plateau Ecology and Agriculture, Qinghai Academy of Animal and Veterinary Sciences, Qinghai University, Xining 810016, China; xiuzhang11@163.com (X.-Z.L.); yulingli2000@163.com (Y.-L.L.)

**Keywords:** heteromorphic stereostructures of MAT1-1-1 and MAT1-2-1 proteins, Bayesian clustering, AlphaFold-predicted 3D protein structures, *Hirsutella sinensis*, reproduction of *Ophiocordyceps sinensis*, sexual life of the natural *Cordyceps sinensis* insect–fungal complex

## Abstract

The MAT1-1-1 and MAT1-2-1 proteins are essential for the sexual reproduction of *Ophiocordyceps sinensis*. Although *Hirsutella sinensis* has been postulated to be the sole anamorph of *O. sinensis* and to undergo self-fertilization under homothallism or pseudohomothallism, little is known about the three-dimensional (3D) structures of the mating proteins in the natural *Cordyceps sinensis* insect–fungal complex, which is a valuable therapeutic agent in traditional Chinese medicine. However, the alternative splicing and differential occurrence and translation of the *MAT1-1-1* and *MAT1-2-1* genes have been revealed in *H. sinensis*, negating the self-fertilization hypothesis but rather suggesting the occurrence of self-sterility under heterothallic or hybrid outcrossing. In this study, the MAT1-1-1 and MAT1-2-1 proteins in 173 *H. sinensis* strains and wild-type *C. sinensis* isolates were clustered into six and five clades in the Bayesian clustering trees and belonged to 24 and 21 diverse AlphaFold-predicted 3D structural morphs, respectively. Over three-quarters of the strains/isolates contained either MAT1-1-1 or MAT1-2-1 proteins but not both. The diversity of the heteromorphic 3D structures of the mating proteins suggested functional alterations of the proteins and provided additional evidence supporting the self-sterility hypothesis under heterothallism and hybridization for *H. sinensis*, Genotype #1 of the 17 genome-independent *O. sinensis* genotypes. The heteromorphic stereostructures and mutations of the MAT1-1-1 and MAT1-2-1 proteins in the wild-type *C. sinensis* isolates and natural *C. sinensis* insect–fungi complex suggest that there are various sources of the mating proteins produced by two or more cooccurring heterospecific fungal species in natural *C. sinensis* that have been discovered in mycobiotic, molecular, metagenomic, and metatranscriptomic studies, which may inspire future studies on the biochemistry of mating and pheromone receptor proteins and the reproductive physiology of *O. sinensis*.

## 1. Introduction

The natural *Cordyceps sinensis* insect–fungal complex is one of the most expensive therapeutic agents in traditional Chinese medicine and has a rich history of clinical applications for centuries in health maintenance, disease amelioration, post-illness and post-surgery recovery, and antiaging therapy [Zhu et al., 1998 [1], 2011 [2]]. As defined by the *Chinese Pharmacopoeia*, natural *C. sinensis* is an insect–fungal complex containing the *Ophiocordyceps sinensis* fruiting body and the remains of a *Hepialidae* moth larva (an intact, thick larval body wall with numerous bristles, an intact larval intestine and head tissues, and fragments of other larval tissues) [Ren et al., 2013 [3]; Zhang et al., 2014 [4]; Lu et al., 2016 [5]; Li et al., 2022 [6], 2023 [7]]. Studies of natural *C. sinensis* have demonstrated its multicellular heterokaryotic structures of hyphal and ascosporic cells and genetic heterogeneity, including at least 17 genomically independent genotypes of *O. sinensis* fungi and >90 other fungal species spanning at least 37 fungal genera and larval genes [Jiang & Yao 2003 [8]; Zhang et al., 2010 [9], 2018 [10]; Xia et al., 2015 [11]; Guo et al., 2017 [12]; Li et al., 2016 [13], 2020 [14], 2022 [6], 2023 [7], 2023 [15]; Zhong et al., 2018 [16]; Kang et al., 2024 [17]]. Among the numerous heterogeneous fungal species, *Hirsutella sinensis* was postulated by Wei et al., 2006 [18] to be the sole anamorph of *O. sinensis*; however, 10 years later, the key authors reported a species contradiction in an artificial cultivation project conducted in a product-oriented industrial setting between anamorphic inoculates of three GC-biased *H. sinensis* strains on *Hepialidae* moth larvae and the sole AT-biased teleomorph (Genotype #4 of *O. sinensis*) in cultivated *C. sinensis* [Wei et al., 2016 [19]]. Notably, the Latin name *Cordyceps sinensis* has been used indiscriminately since the 1840s for both the teleomorph/holomorph of the fungus *C. sinensis* and the wild insect–fungal complex, and the fungus was renamed *Ophiocordyceps sinensis* in 2007 [Sung et al., 2007 [20]; Zhang et al., 2012 [21]; Ren et al., 2013 [3]; Yao & Zhu 2016 [22]; Li et al., 2022 [6]]. Zhang et al., 2013 [23] proposed improper implementation of the “One Fungus=One Name” nomenclature rule of the International Mycological Association [Hawksworth et al., 2011 [24]] while disregarding the presence of multiple genomically independent genotypes of *O. sinensis* fungi and inappropriately replacing the anamorphic name *H. sinensis* with the teleomorphic name *O. sinensis*. Thus, we continue using the anamorphic name *H. sinensis* for Genotype #1 of the 17 *O. sinensis* genotypes in this paper and refer to the genomically independent Genotypes #2–17 fungi as *O. sinensis* before their systematic positions are taxonomically determined, regardless of whether they are genetically GC- or AT-biased. We continue the customary use of the name *C. sinensis* to refer to the wild or cultivated insect–fungal complex because the renaming of *C. sinensis* to *O. sinensis* in 2007 did not involve the indiscriminate use of the Latin name for the natural insect–fungal complex, although this practice will likely be replaced in the future by the differential use of proprietary and exclusive Latin names for the multiple genome-independent *O. sinensis* genotypic fungi and the insect–fungi complex.

The sexual reproductive behavior of ascomycetes is controlled by transcription factors encoded at the mating-type (MAT) locus [Debuchy et al., 2006 [25]; Jones & Bennett 2011 [26]; Zheng & Wang 2013 [27]; Wilson et al., 2015 [28]]. Holliday et al., 2008 [29], Stone et al., 2010 [30], and Hu et al., 2013 [31] reported failures when trying to induce the development of *C. sinensis* fruiting bodies and ascospores via the use of pure *H. sinensis* cultures as inoculants. Zhang et al., 2013 [23] summarized the failures over 40 years of academic experience in research-oriented academic settings. Hu et al., 2013 [31] and Bushley et al., 2013 [32] hypothesized that *H. sinensis* undergoes self-fertilization under homothallism or pseudohomothallism; however, [Zhang et al., 2009 [33], 2011 [34] and Zhang and Zhang 2015 [35] reported the differential occurrence of the *MAT1-1-1* and *MAT1-2-1* genes in numerous wild-type *C. sinensis* isolates and hypothesized that *O. sinensis* underwent facultative hybridization. Moreover, Li et al., 2023 [36], 2024 [37] reported the alternative splicing, differential occurrence, and differential transcription of mating-type and pheromone receptor genes in *H. sinensis* and natural *C. sinensis*, suggesting the occurrence of self-sterility in *H. sinensis* under heterothallism or hybridization and the demand of sexual partners during the sexual life of the natural *C. sinensis* insect–fungi complex.

Sequences of the *MAT1-1-1* and *MAT1-2-1* genes and proteins of *H. sinensis* are available in the GenBank database, but little is known about the polymorphic stereostructures of the proteins in *H. sinensis* strains and wild-type *C. sinensis* isolates, which are extremely crucial to the sexual reproduction of *O. sinensis* and for the maintenance of the natural ecological population volume of the Level II endangered authentic traditional Chinese medicinal “herb” [China Ministry of Agriculture and Rural Affairs 2021 [38]], which is a natural *C. sinensis* insect–fungi complex. In this work, we analyzed and correlated the statistical clustering of the primary structures and AlphaFold-predicted 3D structural models of the MAT1-1-1 and MAT1-2-1 proteins from 173 *H. sinensis* strains and wild-type *C. sinensis* isolates and correlated the heteromorphic structures of the protein sequences encoded by the genome, transcriptome, and metatranscriptome assemblies of *H. sinensis* and natural *C. sinensis*.

## 2. Materials and Methods

### 2.1. C. sinensis Isolates and Accession Numbers of the MAT1-1-1 and MAT1-2-1 Proteins

The AlphaFold database (Cambridgeshire, UK) lists the accession numbers of the MAT1-1-1 and MAT1-2-1 proteins and the 3D protein structures, which were derived from 173 *H. sinensis* strains and wild-type *C. sinensis* isolates that were collected from various production areas on the Qinghai–Tibet Plateau [Zhang et al., 2009 [33], 2011 [34]; Hu et al., 2013 [31]; Zhang & Zhang 2015 [35]; Tunyasuvunakool et al., 2021 [39]].

### 2.2. Genome, Transcriptome, and Metatranscriptome Assemblies of H. sinensis Strains and the Natural C. sinensis Insect–Fungal Complex

The genome assemblies ANOV00000000, JAAVMX000000000, LKHE00000000, LWBQ00000000, and NGJJ00000000 of the *H. sinensis* strains Co18, IOZ07, 1229, ZJB12195, and CC1406-20395, respectively, were used for mating protein analysis [Hu et al., 2013 [31]; Li et al., 2016 [40]; Jin et al., 2020 [41]; Liu et al., 2020 [42]; Shu et al., 2020 [43]].

The transcriptome assembly GCQL00000000 for the *H. sinensis* strain L0106 and the metatranscriptome assembly GAGW00000000 for the natural *C. sinensis* samples collected from Kangding County, Sichuan Province, China, were also used for mating protein analysis [Liu et al., 2015 [44]; Xiang et al., 2014 [45]].

Another metatranscriptome assembly was derived from mature natural *C. sinensis* samples collected from Deqin, Yunnan Province, China (*cf*. the Appendix of [Xia et al., 2017 [46]]). The metatranscriptome assembly sequences were uploaded to a repository database, www.plantkingdomgdb.com/Ophiocordyceps_sinensis/data/cds/Ophiocordyceps_sinensis_CDS.fas (accessed from 18 May 2017 to 18 January 2018), which is currently inaccessible, but a previously downloaded cDNA file was used for mating protein analysis.

### 2.3. Statistical Clustering Analysis for the MAT1-1-1 and MAT1-2-1 Protein Sequences

Multiple protein sequences of the *H. sinensis* strains and wild-type *C. sinensis* isolates were analyzed via the auto mode of MAFFT (v7.427). Bayesian clustering trees of the MAT1-1-1 and MAT1-2-1 protein sequences were then inferred via MrBayes v3.2.7 software (Markov chain Monte Carlo [MCMC] algorithm, New York, NY, USA) with a sampling frequency of 100 iterations after discarding the initial 25% of the samples from a total of 1 million iterations [Huelsenbeck & Ronquist 2001 [47]; Ronquist et al., 2012 [48]; Li et al., 2022 [6], 2023 [49], 2024 [37]]. Clustering analysis was conducted at Nanjing Genepioneer Biotechnologies Co. (Nanjing, China).

### 2.4. AlphaFold-Based Prediction of 3D Structures of Mating Proteins

The 3D structures of the MAT1-1-1 and MAT1-2-1 proteins of the 173 *H. sinensis* strains and wild-type *C. sinensis* isolates were computationally predicted from their amino acid sequences via the artificial intelligence (AI)-based machine learning technology AlphaFold (https://alphafold.com/ (Cambridgeshire, UK), accessed from 18 October 2024 to 31 December 2024) and downloaded from the AlphaFold database for structural polymorphism analysis [Jumper et al., 2021 [50]; David et al., 2022 [51]; Rettie et al., 2023 [52]; Abramson et al., 2024 [53]; Varadi et al., 2024 [54]]. The heteromorphic 3D structures of the MAT1-1-1 and MAT1-2-1 proteins were grouped based on the results of AlphaFold structural and Bayesian clustering analyses.

The AlphaFold database provides per-residue model confidence, the prediction of its score in the local distance difference test (pLDDT), between 0 and 100, a per-residue score that is assigned to each individual residue [Mariani et al., 2013 [55]; Jumper et al., 2021 [50]; David et al., 2022 [51]; Monzon et al., 2022 [56]; Xu et al., 2023 [57]; Abramson et al., 2024 [53]; Varadi et al., 2024 [54]]. Model confidence bands are used to color-code the residues in the 3D structure: 
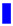
 very high confidence (pLDDT > 90) residues are shown in dark blue, 
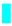
 high (90 > pLDDT > 70) in light blue, 
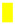
 low (70 > pLDDT > 50) in yellow, and 
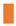
 very low (pLDDT < 50) in orange [Mariani et al., 2013 [55]; Wroblewski & Kmiecik 2024 [58]]. Note that a protein region that is assigned a low pLDDT score does not necessarily indicate that this region is the most variable region in the protein sequence; in contrast, a substantially variable region of a protein may be assigned a high pLDDT score. The AlphaFold database provides an average pLDDT score for each of the predicted 3D structure models of mating proteins, representing the overall model confidence in the predicted 3D structures.

### 2.5. Alignment Analysis of Protein Sequences

The amino acid sequences of the MAT1-1-1 and MAT1-2-1 proteins of *H. sinensis* and natural *C. sinensis* were aligned and compared via the GenBank Blastp program (https://blast.ncbi.nlm.nih.gov/ (Bethesda, MD, USA), accessed from 18 October 2024 to 1 December 2024).

### 2.6. Amino Acid Properties and Scale Analysis

The amino acid components of the mating proteins were scaled based on the general chemical characteristics of their side chains (*cf*. Appendix A) and plotted sequentially with a window size of 21 amino acid residues for the α-helices, β-sheets, β-turns, and coils of the MAT1-1-1 and MAT1-2-1 proteins via the linear weight variation model of the ExPASy ProtScale algorithm (https://web.expasy.org/protscale/ (Basel, Switzerl;and), accessed from 18 October 2024 to 1 December 2024) [Deleage & Roux 1987 [59]; Gasteiger et al., 2005 [60]; Peters & Elofsson 2014 [61]; Simm et al., 2016 [62]; Li et al., 2024b [37]]. The plotting topologies and waveforms of the ProtScale plots for the proteins were compared to explore alterations in the 2D structures of the mating proteins.

## 3. Results

### 3.1. Diversity of the MAT1-1-1 and MAT1-2-1 Proteins in H. sinensis Strains and Wild-Type C. sinensis Isolates on the Basis of the AlphaFold-Predicted 3D Structures

A prior publication [Li et al., 2023 [36], 2024 [37]] reported the differential occurrence, alternative splicing, and differential transcription of mating-type genes in *H. sinensis* and natural *C. sinensis*. The current paper focuses on the diverse stereostructures of the translation products, namely, the MAT1-1-1 and MAT1-2-1 proteins of *O. sinensis*.

The AlphaFold database lists the accession numbers for 138 MAT1-1-1 proteins and 79 MAT1-2-1 proteins, which were derived from 173 *H. sinensis* strains and wild-type *C. sinensis* isolates [Zhang et al., 2009 [33], 2011 [34], 2013 [23]; Bushley et al., 2013 [32]; Hu et al., 2013 [31]; Zhang & Zhang 2015 [35]]. Among the 173 strains/isolates, 42 (24.3%) had records of AlphaFold-predicted 3D structures for both the MAT1-1-1 and MAT1-2-1 proteins. A majority (75.7%) of the strains/isolates presented 3D structure records for either the MAT1-1-1 or MAT1-2-1 protein, suggesting differential cooccurrences of the two mating proteins essential for the sexual reproduction of *O. sinensis*. In addition, strains CS68-2-1229 and CS2 have duplicated accession numbers for either the MAT1-1-1 or MAT1-2-1 protein, unlike the 171 other strains/isolates (i.e., 173 = 138 + 79 – 42 − 2).

Strain CS68-2-1229 has two accession numbers for the MAT1-1-1 protein, namely, AGW27560 and AGW27528, which share 100% sequence identity. However, AGW27560 is a full-length protein containing 372 amino acids, whereas AGW27528 is an N- and C-terminally truncated protein containing 301 amino acids with 80.9% query coverage.

Strain CS2 has two accession numbers for the full-length MAT1-2-1 protein, namely, AEH27625 and ACV60400, which contain 249 amino acids and share 100% sequence identity; however, the sequences were released by GenBank 6 years apart, on 03-JUN-2010 and 25-JUL-2016, respectively.

The 138 MAT1-1-1 proteins belong to diverse 3D structural models or morphs under 24 UniProt codes in the AlphaFold database, 118 of which are full-length proteins belonging to 3D structural morphs under 15 AlphaFold UniProt codes and are listed in Table 1. Among the 118 full-length proteins, 89 (75.4%) are under the UniProt code U3N942 and are considered “likely authentic” proteins. The remaining 20 of the 138 MAT1-1-1 proteins are truncated and belong to 3D structural morphs under nine other UniProt codes.

The 79 MAT1-2-1 proteins belong to diverse 3D structural morphs under 21 UniProt codes in the AlphaFold database, 74 of which are full-length proteins containing 249 amino acids belonging to 3D structural morphs under 17 AlphaFold UniProt codes and are listed in Table 2. Among the 74 full-length MAT1-2-1 proteins, 38 (51.4%) are under the UniProt code D7F2E9 and are considered “likely authentic” proteins. The remaining 5 of the 79 MAT1-2-1 proteins are truncated and belong to 3D structural morphs under four other UniProt codes.

### 3.2. Bayesian Analysis of the MAT1-1-1 and MAT1-2-1 Proteins

Figure 1 shows the Bayesian clustering tree for 40 protein sequences covering the diverse structural morphs of MAT1-1-1 proteins under 24 UniProt codes. The sequences ALH24945, ALH24947, and AGW27560 represent a group of 89 sequences under UniProt code U3N942, which were clustered into Branch A1 of Cluster A, as shown in red alongside the tree in Figure 1. Branch A1 in Figure 1 also includes the full-length MAT1-1-1 proteins under UniProt codes A0A0N9QMM1 and T5A511. Cluster A includes other full-length MAT1-1-1 protein sequences with very similar 3D structures, which were clustered into Branch A2 in pink and Branch A3 in purple alongside the tree. The full-length MAT1-1-1 protein sequences with significantly variable 3D structures were clustered within Clusters B–E in Figure 1, either branched or unbranched, under various UniProt codes in red for Branch 1, in pink for Branch 2, in purple for Branch 3, or in brown for Branch 4.

Many truncated MAT1-1-1 proteins were found under UniProt codes U3N919, U3N6U0, U3N9T9, U3N7G5, U3N6U4, U3NE87, U3N6U8, and U3N7H7, which are shown in green alongside the tree in Figure 1 and were clustered into Branches A1–A3 of Cluster A. In addition, Cluster F contains the truncated MAT1-1-1 proteins under the UniProt code U3NE79 in green alongside the tree in Figure 1.

The 79 MAT1-2-1 proteins have various 3D structural morphs under 21 UniProt codes in the AlphaFold database (*cf*. Table 2), among which 32 representative sequences were subjected to Bayesian clustering analysis, as shown in Figure 2.

Among a total of 79 MAT1-2-1 proteins, 74 are full-length proteins, containing 249 amino acids and belonging to diverse 3D structural morphs under 17 AlphaFold UniProt codes. The remaining five MAT1-2-1 proteins are truncated and belong to 3D structural morphs under four other UniProt codes. Among the 74 full-length MAT1-2-1 proteins, 39 (52.7%) proteins under the UniProt codes D7F2E9 and T5AF56 were clustered into Branch I-1 of Cluster I of the Bayesian tree shown in Figure 2 and Table 2. Branch I-2 of Cluster I includes three MAT1-2-1 protein sequences with very similar 3D structures belonging to 3D structural morphs under the UniProt codes V9LW10, D7F2H1, and D7F2F2, as shown in Figure 2. Twenty-seven other full-length MAT1-2-1 proteins with significantly variable 3D structures were within Clusters II–V under various UniProt codes. Branches V-1 and V-2 of Cluster V also include five truncated MAT1-2-1 proteins under different UniProt codes, which are shown in green alongside the tree.

The GenBank database contains five other MAT1-2-1 protein sequences, namely, AFX66471, AFX66481, AFX66483, AFX66485, and AFX66486, which were derived from wild-type *C. sinensis* isolates YN09_3, YN09_96, YN09_140, NP10_1, and NP10_2, respectively, with predicted 3D structure records in the AlphaFold database for the MAT1-1-1 proteins but not for the MAT1-2-1 proteins (*cf*. Table 1 and Table 2). The five MAT1-2-1 protein sequences are 100% identical to the reference sequence ACV60363 of Branch V-1 in the Bayesian clustering tree (*cf*. Figure 2 and Appendix A), indicating that the five protein sequences likely belong to Branch V-1 of Cluster V, together with five other Branch V-1 proteins, including the reference protein ACV60363, as listed in Table 1 and Figure 2.

### 3.3. Heteromorphic AlphaFold-Predicted 3D Structures of the MAT1-1-1 Proteins

Figure 3 shows the AlphaFold-predicted 3D structures of the 118 full-length MAT1-1-1 proteins under 15 structural morphs (Panels A–O), which are also listed in Table 1. Among the 118 full-length proteins, 89 (75.4%) are under the UniProt code U3N942, as predicted by AlphaFold technology (Panel A of Figure 3). This 3D structure model most likely represents the authentic protein structure with full mating functionality.

As shown in Table 1 and Figure 1 and Figure 3, 94 (79.9%) of the 118 full-length MAT1-1-1 proteins are under the UniProt codes U3N942, A0A0N9QMM1, and T5A511 and clustered into Branch A1 of Cluster A in the Bayesian tree belonging to the 3D structure morphs A–C. The 94 full-length MAT1-1-1 proteins are most frequently detected and are likely authentic with full mating functionality.

Figure 4 shows the sequence distributions (Panel A) and predicted 3D structures of 20 other MAT1-1-1 proteins, which are truncated at the N- and C-termini; these structures constitute the remaining nine diverse morphs of 3D structures (Panels B–J).

### 3.4. Heteromorphic AlphaFold-Predicted 3D Structures of the MAT1-2-1 Proteins

Among the 79 MAT1-2-1 protein sequences belonging to the 21 diverse 3D structural morphs, 74 are full-length proteins, 69 of which belong to diverse structural morphs under 17 UniProt codes and are shown in Panels A–Q of Figure 5.

As shown in Table 2 and Figure 2 and Figure 5, 39 (52.7%) of the 74 full-length MAT1-2-1 proteins are under the UniProt codes D7F2E9 and T5AF56 and clustered into Branch I-1 of Cluster I in the Bayesian tree belonging to 3D structural morph A of the MAT1-2-1 proteins. The 39 full-length MAT1-2-1 proteins are frequently detected and are likely authentic with full mating functionality.

As shown in Appendix A, the five other full-length MAT1-2-1 protein sequences (AFX66471, AFX66481, AFX66483, AFX66485, and AFX66486) exhibit 100% sequence identity with the Branch V-1 reference protein ACV60363. Thus, the five protein sequences without AlphaFold-predicted 3D structure records likely belong to the 3D structural morph K, together with the Branch V-1 reference protein ACV60363 (*cf*. Figure 2 and Figure 5).

Figure 6 shows the sequence distribution (Panel A) and the AlphaFold-predicted 3D structures of the C-terminally truncated MAT1-2-1 proteins, which constitute the remaining four diverse morphs of 3D structures (Panels B–E).

### 3.5. Primary Structures of the MAT1-1-1 Proteins

Because of the diversity of the 3D structures of the MAT1-1-1 proteins, the variations in their primary amino acid sequences were then analyzed. The 118 full-length MAT1-1-1 proteins (*cf*. Table 1) consisted of 372 amino acids and contributed to 15 diverse 3D structural morphs (*cf*. Figure 3). Among the 118 full-length proteins, 89 shared 100% sequence identity with the query protein sequence (AGW27560), whereas 20 other proteins shared 98.1–99.6% sequence similarity with the query sequence, as they contained various conservative and nonconservative substitutions of amino acid residues at isolated sites, which may have an impact on mating function. Figure 7 shows the alignment of the full-length MAT1-1-1 protein sequences covering five Bayesian clusters, A–E (Branches A1, A2, A3, B, C, D1, D2, E1, E2, E3, and E4), as shown in Figure 1 and Table 1, and 15 AlphaFold 3D structural morphs, A–O (*cf*. Figure 3), as well as the MAT1-1-1 protein sequences encoded by the genome assemblies of *H. sinensis* and the metatranscriptome assemblies of the natural *C. sinensis* insect–fungal complex.

The MAT1-1-1 protein contains a MATalpha_HMGbox domain, which is found in high-mobility group (HMG) proteins involved in DNA binding [Hu et al., 2013 [31]]. This domain is located in segment 51→225 of the query sequence AGW27560, as highlighted in blue and underlined in Figure 7. Some nonconservative residue substitutions in other MAT1-1-1 proteins were found within this domain, as shown in red in Table 3.

The full-length MAT1-1-1 protein sequence EQK97643 (372 aa) was derived from *H. sinensis* strain Co18 under the AlphaFold UniProt T5A511 and published in GenBank on 22-MAR-2015 (*cf*. Table 1; Figure 3). A segment of the genome assembly ANOV01017390 (410←1519), which was also annotated as KE657544 (410←1519) in GenBank, was derived from the same *H. sinensis* strain but published in GenBank on 20-AUG-2013 and was found to be C-terminally truncated (352 aa; 95.1% query coverage vs. EQK97643).

Table 3 summarizes the protein sequence alignment results, including mutant amino acid residues and the percent similarity vs. the sequences of the “likely authentic” full-length MAT1-1-1 protein AGW27560, which are correlated with the statistical and structural analytical results obtained from Bayesian clustering and 3D structure prediction (including the stereostructure morphs and the associated AlphaFold UniProt codes) (*cf*. Table 1, Figure 1, Figure 3 and Figure 7). The correlations shown in Table 3 indicate that the minor sequence differences within or outside the MATalpha_HMGbox domain of the MAT1-1-1 protein sequences may have had an impact on the diverse 3D structures.

Among the 138 MAT1-1-1 protein sequences, 20 are truncated at both the N- and C-termini, showing 68–80% query coverage and belonging to nine diverse 3D structural morphs (*cf*. Figure 4). Eighteen of the twenty truncated proteins presented 100% sequence identity with the representative full-length MAT1-1-1 protein AGW27560 under the UniProt code U3N942 (*cf*. Table 1). Among the 18 truncated proteins, 17 were clustered into Branch A1 of Cluster A in the Bayesian clustering tree (*cf*. Figure 1); however, the protein AGW27526 under the UniProt code U3N7G5 showed the longest Bayesian clustering distance vs. other truncated proteins and was clustered into Branch A3 of Cluster A (*cf*. Figure 1). Two other truncated proteins (AGW27522 and AGW27536) under the UniProt codes U3N6U0 and U3N7H7 shared 99.6–99.7% sequence similarity with the query sequence AGW27560 with either a nonconservative P-to-L substitution or an L residue deletion and were clustered into Branches A2 and A3 of Cluster A, respectively.

Figure 7 also shows the C-terminally truncated MAT1-1-1 proteins encoded by the genome assemblies ANOV01017390, LKHE01001116, and JAAVMX010000001 of *H. sinensis* strains Co18, 1229, and IOZ07, respectively [Hu et al., 2013 [31]; Li et al., 2016 [40], 2024 [37]; Shu et al., 2020 [43]]. The truncated MAT1-1-1 proteins encoded by the genome assemblies had a deletion of 19 amino acid residues (SHLPPSPPHNPLDDFYFAF) at the *C-termini* and contained a nonconservative T-to-S substitution (*cf*. Figure 7). The MAT1-1-1 protein encoded by the metatranscriptome assembly GAGW01008880 of natural *C. sinensis* is truncated by 96 amino acids at the N-terminus (MTTRNEVMQRLSSVRADVLLNFLTDDAIFQLA-SRHESTTEADVLTPVSTAAASRATRQTKEASCDRAKRPLNAFMAFRSYYLKLPDVQQ-QKTASG) partially with and outside the MATalpha_HMGbox domain [Hu et al., 2013 [31]; Xiang et al., 2014 [45]; Li et al., 2024 [37]]. The MAT1-1-1 protein encoded by the metatranscriptome assembly OSIN7648 features midsequence truncation with a deletion of 18 amino acids (SMQREYQAPRFFYDYSVS) outside the MATalpha_HMGbox domain and a nonconservative L-to-F substitution within the exon II-encoding region of the *MAT1-1-1* gene [Xia et al., 2017 [46]; Li et al., 2024 [37]].

### 3.6. Primary Structures of the MAT1-2-1 Proteins

Among the 74 MAT1-2-1 proteins available in the AlphaFold database, 69 are full-length proteins containing 249 amino acids and are attributed to diverse 3D structural morphs under 17 UniProt codes (*cf*. Figure 5; Table 2). Among the 69 full-length proteins, 39 are 100% identical to the query sequence AEH27625 and clustered into Branch I-1 of Cluster I of the Bayesian tree (*cf*. Figure 2). The remaining 30 full-length proteins share 97.6–99.6% sequence similarity with the query protein sequence, containing various conservative and nonconservative substitutions of amino acid residues at isolated sites, which may have an impact on the mating function.

Figure 8 shows the alignment of the full-length MAT1-2-1 protein sequences covering five Bayesian clusters (Branches I-1, I-2, II-1, II-2, III, IV-1, IV-2, V-1, and V-2; *cf*. Figure 2) and 17 AlphaFold 3D structural morphs (*cf*. Figure 5 and Table 2), as well as the MAT1-2-1 protein sequences encoded by the genome and transcriptome assemblies of *H. sinensis* and the metatranscriptome assembly of the natural *C. sinensis* sample that was collected from Deqin, Yunnan Province, China. According to GenBank, both the MAT1-2-1 protein sequence EQL04085 and the genome assembly ANOV01000063 (9329→10,182) were derived from *H. sinensis* strain Co18 and submitted to GenBank by the same group of authors. However, the genome assembly ANOV01000063 (9329→10,182) contains a conservative S-to-A substitution, whereas EQL04085 does not contain this substitution. Note: the arrows “→” and “←” indicate sequences in the sense and antisense strands of the genomes, respectively.

The MAT1-2-1 protein contains an HMG-box_ROX1-like domain, which binds the minor groove of DNA in a sequence-specific manner [Hu et al., 2013 [31]]. This domain is located in segment 127→197 of the query sequence AEH27625, which is shown in blue and underlined in Figure 8. Some conservative residue substitutions in other MAT1-2-1 proteins were found within this domain, as shown in red in Table 4.

Table 4 summarizes the protein sequence alignment results, including mutant amino acid residues and percent similarities relative to the “likely authentic” full-length MAT1-2-1 protein AEH27625. The sequence alignment results are correlated with the statistical and structural analytical results obtained from the Bayesian clustering and 3D structure prediction (including the stereostructure models and the associated AlphaFold UniProt codes) (*cf*. Table 2, Figure 2, Figure 5 and Figure 8). The correlations shown in Table 4 indicate that the minor sequence differences within or outside the HMG-box_ROX1-like domain of the MAT1-2-1 protein sequences may have an impact on the diverse 3D structures.

In addition to the 74 full-length MAT1-2-1 proteins, five other protein sequences are C-terminally truncated and were clustered into Branches V-1 and V-2 of Cluster V in the Bayesian tree (shown in green in Figure 2), exhibiting 69–74% query coverage and 99.4–99.5% protein sequence similarity to the reference full-length MAT1-2-1 protein AEH27625 under the UniProt code D7F2E9. The five truncated MAT1-2-1 proteins contain a single conserved Y-to-H substitution with the HMG-box_ROX1-like domain, belonging to different 3D structural morphs under the 4 AlphaFold UniProt codes (*cf*. Figure 6).

Figure 8 also shows a conservative S-to-A substitution with the HMG-box_ROX1-like domain in the MAT1-2-1 proteins encoded by the genome assemblies ANOV01000063, LKHE01001605, LWBQ01000021, and NGJJ01000619 of *H. sinensis* strains Co18, 1229, ZJB12195, and CC1406-20395, respectively [Hu et al., 2013 [31]; Li et al., 2016 [40], 2023 [36], 2024 [37]; Jin et al., 2020 [41]; Liu et al., 2020 [42]]. A conservative Y-to-H substitution with the HMG-box_ROX1-like domain was found in the transcriptome assembly GCQL01020543 of the *H. sinensis* strain L0106 [Liu et al., 2015 [44]; Li et al., 2024 [37]]. No mutation was detected in the MAT1-2-1 protein encoded by the metatranscriptome assembly OSIN7649 of natural *C. sinensis* [Xia et al., 2017 [46]; Li et al., 2024 [37]], indicating no variation in the 3D structures and mating functionality of the MAT1-2-1 protein.

### 3.7. Differential Genomic Occurrence of the MAT1-1-1 and MAT1-2-1 Proteins in H. sinensis

Table 5 lists the differential occurrence of the MAT1-1-1 and MAT1-2-1 proteins encoded by the genome assemblies ANOV01017390/ANOV01000063, JAAVMX010000001, LKHE01001116/LKHE01001605, LWBQ01000021, and NGJJ01000619 of the *H. sinensis* strains Co18, IOZ07, 1229, ZJB12195, and CC1406-20395, respectively [Hu et al., 2013 [31]; Li et al., 2016a [40]; Jin et al., 2020 [41]; Liu et al., 2020 [42]; Shu et al., 2020 [43]]. The genome assemblies LWBQ00000000 and NGJJ00000000 of the *H. sinensis* strains do not contain the genes encoding the MAT1-1-1 proteins, and the genome assembly JAAVMX000000000 does not contain the gene encoding the MAT1-2-1 protein.

Although repetitive copies of many genes have been identified in the *H. sinensis* genome assemblies and the mutant repetitive genome sequences have normal transcriptional abilities and encode mutant proteins with altered functional specificities [Li et al., 2024 [63]], no repetitive copies of the *MAT1-1-1* or *MAT1-2-1* genes were identified in the *H. sinensis* genome assemblies, proving that the mutant mating proteins with diverse stereostructures may be encoded by mutant mating-type genes of certain *H. sinensis* strains or heterospecific fungal species other than *H. sinensis*.

### 3.8. Differential Transcriptomic and Metatranscriptomic Occurrences of the MAT1-1-1 and MAT1-2-1 Proteins in H. sinensis and the Natural C. sinensis Insect–Fungal Complex

Table 6 shows the differential occurrence of the MAT1-1-1 and MAT1-2-1 proteins encoded by the transcriptome assembly of the *H. sinensis* strain L0106 and the metatranscriptome assemblies of natural *C. sinensis* specimens [Xiang et al., 2014 [45]; Liu et al., 2015 [44]; Xia et al., 2017 [46]]. The transcriptome assembly GCQL00000000 does not contain the gene encoding the MAT1-1-1 protein [Liu et al., 2015 [44]], and the metatranscriptome assembly GAGW00000000 of natural *C. sinensis* does not contain the gene encoding the MAT1-2-1 protein [Xiang et al., 2014 [45]]. Worth mentioning, Bushley et al., 2013 [32] and Li et al., 2023 [36], 2024 [37] reported disrupted translation of the MAT1-2-1 transcript due to alternative splicing of the gene with unspliced intron I, which contains three stop codons, in *H. sinensis* strain 1229. The alternative splicing of the *MAT1-2-1* gene eventually produces a largely truncated protein lacking the C-terminal portions (including the entire HMG-box_ROX1-like domain) of the protein encoded by exons II and III of the gene and significantly alters the 3D structure of the protein with dysfunctionality.

### 3.9. Diverse Secondary (2D) Structures of the MAT1-1-1 Proteins Encoded by the Genome of H. sinensis and the Metatranscriptome of Natural C. sinensis Insect–Fungal Complex

The predicted 3D structures of the truncated MAT1-1-1 proteins encoded by the genome and metatranscriptome assemblies are not available in the AlphaFold database. Figure 9 shows changes in the 2D structures, α-helices (Panel A), β-sheets (Panel B), β-turns (Panel C), and coils (Panel D) of the truncated MAT1-1-1 proteins that were analyzed via ExPASy ProtScale technology. EQK97643 (372 aa) of the *H. sinensis* strain GS09_111 was used as the reference for the authentic MAT1-1-1 proteins for the 2D analysis shown in the upper plots in all panels of Figure 9, which was clustered into Branch A1 of Cluster A (*cf*. Figure 1) and belongs to 3D structural morph A under the UniProt code U3N942 (*cf*. Figure 3).

The upper-middle plots in all panels show the 2D structures of the MAT1-1-1 protein encoded by the genome assembly ANOV01017390 of the *H. sinensis* strain Co18, which also represents two other MAT1-1-1 sequences within the genome assemblies LKHE01001116 and JAAVMX010000001 of the *H. sinensis* strains 1229 and IOZ07, respectively. The lower-middle and lowest plots in each panel of Figure 9 present the 2D structures of the MAT1-1-1 proteins encoded by the metatranscriptome assemblies GAGW01008880 and OSIN7648 of natural *C. sinensis*, respectively.

The open boxes shown in red in the EQK97643 plots indicate the C-terminal truncation region occurring in the genome assembly ANOV01017390 of *H. sinensis* strain Co18, which also represents two other genome assemblies LKHE01001116 and JAAVMX010000001 of *H. sinensis* strains 1229 and IOZ07, respectively.

The N-terminal truncation region of the MAT1-1-1 protein encoded by the metatranscriptome assembly GAGW01008880 of natural *C. sinensis* is indicated with open boxes shown in blue in the reference protein EQK97643 plots in Figure 9.

The open boxes shown in green in the OSIN7648 plots, as well as the corresponding region in the reference EQK97643 plots for the authentic MAT1-1-1 protein for structural comparison, outline the changes in the topology and waveform of the α-helix, β-sheet, β-turn, and coil in the midsequence truncation region in the MAT1-1-1 protein encoded by the metatranscriptome assembly OSIN7648. The topology and waveform changes appear to be more dramatic in the α-helix and β-turn plots in the midsequence truncation region of the protein OSIN7648 than in the β-sheet and coil plots. The midsequence truncation and apparent changes in the 2D structures imply significant alterations in the variable protein folding and 3D structures of the MAT1-1-1 proteins encoded by the genome and metatranscriptome assemblies and their mating functionalities.

### 3.10. Diverse 2D Structures of the MAT1-2-1 Proteins Encoded by the Genomes, Transcriptomes, and Metatranscriptomes of H. sinensis and the Natural C. sinensis Insect–Fungal Complex

The predicted 3D structures of the mutant MAT1-2-1 proteins encoded by these genome, transcriptome, and metatranscriptome assembly sequences are unavailable in the AlphaFold database. Figure 10 shows the 2D structures of the MAT1-2-1 proteins for the α-helices (Panel A), β-sheets (Panel B), β-turns (Panel C), and coils (Panel D). Each panel of Figure 10 contains two ProtScale plots for two MAT1-2-1 proteins. AEH27625 (249 aa) of the *H. sinensis* strain CS2 was used as the reference for the full-length MAT1-2-1 protein shown in the upper plots of all panels. The lower plots in all panels of Figure 10 represent the MAT1-2-1 protein encoded by the genome assembly ANOV01000063 (9329→10,182), also representing three other genomic sequences, namely, NGJJ01000619 (23,030←23,883), LWBQ01000021 (238,873←239,726), and LKHE01001605 (13,860←14,713). Figure 10 shows slight changes in the topology and waveforms of the α-helices, β-turns, and coils in the MAT1-2-1 protein sequences encoded by the genome assembly ANOV01000063. The variation regions are outlined with the open boxes shown in red in the ANOV01000063 plots in Panels A, C, and D, respectively, as well as in the corresponding region in the reference MAT1-2-1 protein AEH27625 plots for topology and waveform comparisons. No apparent 2D changes were found in the topology or waveforms of the β-sheets of the MAT1-2-1 proteins encoded by the genome sequences.

In addition, no apparent changes were observed in the topology and waveforms in the ProtScale plots for the MAT1-2-1 proteins encoded by the transcriptome assembly GCQL01020543 (397←1143) of *H. sinensis* strain L0106 and the metatranscriptome assembly OSIN7649 (1→249) of natural *C. sinensis*, as compared with those for the reference protein AEH27625 [Liu et al., 2015 [44]; Xia et al., 2017 [46]; Li et al., 2024 [37]]. Thus, these fully functional MAT1-2-1 proteins potentially belong to 3D structural morph A under the UniProt code U3N942 (*cf*. Figure 3) and are included within Branch I-1 of Cluster I in the Bayesian tree (*cf*. Figure 2).

## 4. Discussion

### 4.1. Protein 3D Structure Analysis via the AI-Based AlphaFold Prediction System in Combination with Statistical Bayesian Clustering Technology to Stratify 3D Structure Models

The AI-based AlphaFold 3D structure prediction technology displayed extremely refined and powerful abilities in exploring and uncovering the subtle sequence differences of the proteins and in revealing the impact of these subtle sequence differences on 3D structure predictions. The AlphaFold database contains 15 diverse 3D structural models for the full-length MAT1-1-1 proteins (excluding the truncated proteins) and 17 diverse 3D structural models for the full-length MAT1-2-1 proteins (*cf*. Figure 3 and Figure 5).

As shown in Figure 1, Figure 2, Figure 3, Figure 4, Figure 5 and Figure 6 and Table 1 and Table 2, the complexity of the 3D structure heteromorphs predicted using AlphaFold technology was clarified and organized after the combined use of statistical Bayesian clustering analysis. This combination of the two powerful technologies helped stratify the 3D structural heteromorphic groups. As a result, the 15 and 17 stereostructure heteromorphs for the full-length MAT1-1-1 and MAT1-2-1 proteins, respectively, were stratified into five Bayesian clusters for each of the mating proteins, regardless of whether the clusters were branched or unbranched. Thus, different 3D structure models within a special Bayesian branch are closely structurally related, whereas other models may be structurally and statistically distant.

### 4.2. Heteromorphic 3D Structures of the MAT1-1-1 and MAT1-2-1 Proteins in H. sinensis Strains and Wild-Type C. sinensis Isolates

The study presented in this paper demonstrated the heteromorphic 3D structures of the MAT1-1-1 and MAT1-2-1 proteins in 173 *H. sinensis* strains and wild-type *C. sinensis* isolates. Appropriate interaction of the MAT1-1-1 and MAT1-2-1 proteins with functional stereostructures is essential for the sexual reproduction of *O. sinensis* during the lifecycle of the natural *C. sinensis* insect–fungi complex. However, 75.7% of the strains/isolates contained either MAT1-1-1 or MAT1-2-1 proteins but did not generate corresponding pairing mating proteins. These strains/isolates were harvested from scattered production locations on the Qinghai–Tibet Plateau. The harvesting locations and protein accession information are available in both the GenBank and AlphaFold databases.

A total of 6 Bayesian clusters with clustering branches and 24 AlphaFold-predicted 3D structural morphs were demonstrated for the heteromorphic stereostructures of 138 MAT1-1-1 proteins (*cf*. Figure 1, Figure 3 and Figure 4). The full-length and truncated MAT1-1-1 proteins belonged to 15 and 9 diverse morphs of 3D structures, respectively. The most frequently detected “authentic” MAT1-1-1 proteins were clustered into Branch A1 of Cluster A in the Bayesian tree (*cf*. Figure 1 and Table 1), belonging to 3D structural morph A of the MAT1-1-1 proteins (*cf*. Figure 3). Figure 7 and Table 3 show that the scattered amino acid residue substitutions, conservative or nonconservative, are located within or outside the MATalpha_HMGbox domain of MAT1-1-1 protein sequences [Hu et al., 2013 [31]].

A total of five Bayesian clusters with clustering branches and 21 AlphaFold 3D structural morphs were demonstrated for the heteromorphic stereostructures of 79 MAT1-2-1 proteins (*cf*. Figure 2, Figure 5 and Figure 6). The full-length and truncated MAT1-2-1 proteins belonged to 17 and 4 diverse morphs of 3D structures, respectively. Many (52.7%) of the MAT1-2-1 proteins were clustered into Branch I-1 of Cluster I in the Bayesian tree (*cf*. Figure 2 and Table 2), belonging to 3D structural morph A of the MAT1-2-1 proteins (*cf*. Figure 5). Figure 8 and Table 4 show that the dispersed amino acid residue substitutions, conservative or nonconservative, are located within or outside the HMG-box_ROX1-like domain of MAT1-2-1 protein sequences [Hu et al., 2013 [31]].

Zhang and Zhang [2015 [35]] reported 4.7% and 5.7% variations in the exon sequences of the *MAT1-1-1* and *MAT1-2-1* genes, respectively, in numerous wild-type *C. sinensis* isolates. Exon variations may disrupt the translation of coding sequences, in addition to alternative splicing and differential occurrence and transcription of mating genes, as reported by Li et al., 2023 [36], 2024 [37]. The mutation rates reported by Zhang and Zhang 2015 [35] appeared to be much greater than those present in the GenBank database, which includes numerous variable sequences of the MAT1-1-1 and MAT1-2-1 proteins of *C. sinensis* isolates [Li et al., 2024 [37]]. Presumably, Zhang and Zhang [2015 [35]] might not have uploaded all the variable sequences to the GenBank database to truthfully represent the natural diversity of the variable stereostructures of the mating proteins.

Although the AlphaFold database does not include the predicted 3D structures for the mating proteins encoded by the genome and transcriptome assemblies of *H. sinensis* strains and the metatranscriptome assemblies of the natural *C. sinensis* insect–fungi complexes, the 2D structures of the mutant MAT1-1-1 and MAT1-2-1 proteins (including those with truncation of large protein segments) that were translated from the nucleotide sequences were analyzed to explore the variations in the α-helices, β-sheets, β-turns, and coils via ExPASy ProtScale technology (*cf*. Figure 7, Figure 8, Figure 9 and Figure 10). The apparent changes in the 2D structures of the mating proteins indicate altered 3D structures and subsequent dysfunction and even complete deactivation of the mating proteins.

Research has confirmed that the MAT1-1-1 and MAT1-2-1 proteins can interact with each other [Jacobsen et al., 2002 [64]; Rams & Kück 2022 [65]]. The interaction of mating proteins with functional stereostructures is essential for the recognition of compatible mating partners and plays a crucial role in regulating sexual reproduction by controlling gene expression related to mating compatibility within a fungal species. The MAT1-1-1 protein contains a MATalpha_HMGbox domain, and the MAT1-2-1 protein contains an HMG-box_ROX1-like domain [Hu et al., 2013 [31]]. Both domains are involved in DNA binding, binding to specific DNA sequences in the genome to regulate the transcription of genes involved in mating processes. We highlighted the domains in the sequences of the mating proteins in Figure 7 and Figure 8 and summarized the conservative and nonconservative amino acid residue substitutions within or outside the domains in the sequences of the mating proteins of the wild-type isolates in Table 3 and Table 4.

The heteromorphic stereostructures of the mating proteins might explain, at least partially, why efforts made in the past 40–50 years to cultivate pure *H. sinensis*, Genotype #1 of the 17 *O. sinensis* genotypes, in research-oriented academic settings to induce the production of fruiting bodies and ascospores have consistently failed, as reported and summarized previously [Holliday & Cleaver 2008 [29]; Stone 2010 [30]; Hu et al., 2013 [31]; Zhang et al., 2013 [23]]. Table 5 and Table 6 of this paper confirmed the differential occurrence of the mating proteins encoded by the genome and transcriptome assemblies of the *H. sinensis* strains and by the metatranscriptome assemblies of the natural *C. sinensis* insect–fungi complex. Bushley et al., 2013 [32] and Li et al., 2024 [37] reported alternative splicing of the *MAT1-2-1* gene with unspliced intron I, which contains three stop codons, in *H. sinensis* strain 1229. Consequently, the C-terminally truncated MAT1-2-1 protein fragment lacked the major portion of the protein encoded by exons II and III of the gene.

### 4.3. Differential Occurrences of MAT1-1-1 and MAT1-2-1 Proteins with Heteromorphic Sereostructures in H. sinensis Strains and C. sinensis Isolates

Uploaded to the GenBank and AlphaFold databases by many researchers, 131 (75.7%) of the 173 *H. sinensis* strains and wild-type *C. sinensis* isolates generated either the MAT1-1-1 or MAT1-2-1 protein but not both. This phenomenon was confirmed by the differential occurrence of the mating proteins encoded by the genome, transcriptome, and metatranscriptome assemblies of the *H. sinensis* strains and the natural *C. sinensis* insect–fungi complex samples (*cf*. Table 5 and Table 6).

However, 42 other strains/isolates (24.3%) produced both MAT1-1-1 and MAT1-2-1 proteins, the sequences of which were derived from genomic DNA isolated from *H. sinensis* strains or wild-type *C. sinensis* isolates but, unfortunately, not detected directly via gene transcription assays or biochemical examinations. In addition to the intracellular biological processes of the MAT1-1-1 and MAT1-2-1 proteins, such as differential transcription and alternative splicing of the genes, as reported by Li et al., 2024 [37], it needs to be considered whether the mating proteins with heteromorphic 3D structures are capable of expressing their mating functions to accomplish the sexual reproduction of *O. sinensis* during the lifecycle of the natural *C. sinensis* insect–fungi complex.

The MAT1-1-1 proteins from 35 of the 42 strains/isolates were clustered into Bayesian Cluster A (*cf*. Figure 1), accompanied by one of the twenty MAT1-2-1 proteins that were clustered into Bayesian Cluster I or one of the fifteen MAT1-2-1 proteins that were clustered into Bayesian Cluster V (*cf*. Figure 2). These results will be meaningful and useful for the future design of biochemical protein research and reproductive physiological research to explore the functionalities of mating proteins that are clustered into different Bayesian clusters and have diverse 3D structural morphs. The most challenging aspect of biochemical examinations is determining the stereostructures and molecular dynamics of fully functioning proteins under native conditions or after renaturation [Zhu & Gray 1994 [66]].

Figure 7 and Figure 8 show the amino acid residue substitutions in the mating protein sequences of the *H. sinensis* strains and wild-type *C. sinensis* isolates, which cause the alterations of 3D structures shown in Figure 1 and Figure 6 and Table 1 and Table 2. Table 3 summarizes the correlation between the altered primary structures and the diverse stereostructures. Some amino acid residue substitutions occur within the MATalpha_HMGbox domain and HMG-box_ROX1-like domain of the MAT1-1-1 and MAT1-2-1 protein sequences, respectively. In addition to the single-residue substitutions, Figure 7 and Figure 8 show more pronounced mutations in the mating proteins encoded by the genome, transcriptome, and metatranscriptome assemblies of *H. sinensis* strains and natural *C. sinensis* insect–fungi complexes, which caused dramatic changes in the 2D structures of the mating proteins (*cf*. Figure 9 and Figure 10). These findings will inspire further refined research to accurately locate the causal relationship between specific variations in amino acid sequences and protein 3D structural changes, as well as the potential impact on mating protein functionality.

### 4.4. Heteromorphic 3D Structures of the Mating Proteins and Sexual Reproductive Behavior of H. sinensis, Genotype #1 of O. sinensis

Sexual reproduction of *O. sinensis* is crucial for maintaining the natural population volume of the *C. sinensis* insect–fungi complex, which is endangered at Level 2 of National Key Protected Wild Plants [China Ministry of Agriculture and Rural Affairs 2021 [38]]. The following hypotheses have been previously proposed for *H. sinensis*, the sole anamorph of *O. sinensis* postulated by Wei et al., 2006 [18]: (1) homothallism [Hu et al., 2013 [31]], (2) pseudohomothallism [Bushley et al., 2013 [32]], and (3) facultative hybridization [Zhang & Zhang 2015 [35]]. These hypotheses are based on nucleotide data derived from molecular, genome, and transcriptome studies of *H. sinensis*. In theory, self-fertilization under homothallism and pseudohomothallism in ascomycetes becomes a reality when the interaction of MAT1-1-1 and MAT1-2-1 proteins with appropriate stereostructures exhibit full mating functions within a single fungal cell [Turgeon & Yoder 2000 [67]; Debuchy & Turgeo 2006 [25]; Jones & Bennett 2011 [59]; Zhang et al., 2011 [34]; Bushley et al., 2013 [32]; Hu et al., 2013 [31]; Zheng & Wang 2013 [27]; Wilson et al., 2015 [28]; Zhang & Zhang 2015 [35]]. However, after thoroughly analyzing genetic and transcriptional data for *H. sinensis* in the literature, Li et al., 2023 [36], 2024b [37] reported differential occurrences, alternative splicing, and differential transcription of the mating-type genes of the MAT1-1 and MAT1-2 idiomorphs and the pheromone receptor genes in 237 *H. sinensis* strains. Thus, the evidence indicated that *H. sinensis*, which is Genotype #1 of the 17 genomically independent *O. sinensis* genotypes, is self-sterilizing and incapable of completing self-fertilization but requires sexual partners to accomplish *O. sinensis* sexual reproduction under heterothallism or hybridization.

Because of the absence of repetitive copies of the *MAT1-1-1* or *MAT1-2-1* genes in the *H. sinensis* genome assemblies, the alternative splicing and differential occurrence and transcription of the mating-type genes and the diversity of heteromorphic 3D structures of the mating proteins with altered functionalities indicate that there may be two or more *H. sinensis* populations, either monoecious or dioecious. The different *H. sinensis* populations may participate as sexual partners capable of producing either the functioning MAT1-1-1 or the functioning MAT1-2-1 protein with proper stereostructures for reciprocal pairing and interaction during successful physiological heterothallism crossing. Thus, a- and α-pheromones and corresponding α- and a-pheromone receptor proteins play critical roles in the communication of sexual signals between sexual partners. If this assumption is correct, the sexual partners might possess indistinguishable *H. sinensis*-like morphological and growth characteristics, as elucidated previously [Kinjo & Zang 2001 [68]; Zhang et al., 2009 [33]; Chen et al., 2011 [69]; Li et al., 2013 [70], 2016 [13]; Mao et al., 2013 [71]]. For instance, the indistinguishable *H. sinensis* strains 1229 and L0106 produce complementary transcripts of the mating-type genes and proteins of the MAT1-1 and MAT1-2 idiomorphs [Bushley et al., 2013 [32]; Liu et al., 2015 [44]; Li et al., 2023 [36], 2024 [37]].

Even if the physiological heterothallism hypothesis is incorrect for *O. sinensis*, one of the mating proteins might be produced by genome-independent, heterospecific fungal species, which would result in plasmogamy and the formation of heterokaryotic cells (*cf*. Figure 3 of [Bushley et al., 2013 [32]]) to ensure successful sexual hybridization or even parasexual reproduction if the heterospecific species are capable of breaking interspecific reproduction isolation, similar to many cases of fungal sexual hybridization and parasexual reproduction that promote adaptation to the extremely adverse ecological environment on the Qinghai–Tibet Plateau [Pfennig 2007 [72]; Seervai et al., 2013 [73]; Nakamura et al., 2019 [74]; Du et al., 2020 [75]; Hėnault et al., 2020 [76]; Samarasinghe et al., 2020 [77]; Mishra et al., 2021 [78]; Steensels et al., 2021 [79]; Kück et al., 2022 [80]]. Alternatively, to complete physiological heterothallism or hybridization for reproduction, mating partners that produce the MAT1-1-1 and MAT1-2-1 proteins with functional stereostructures might exist in adjacent hyphal cells, which might determine their mating choices, and they may communicate with each other through a mating signal-based transduction system of pheromones and pheromone receptors and form “H”-shaped crossings of multicellular hyphae [Hu et al., 2013 [31]; Bushley et al., 2013 [32]; Mao et al., 2013 [71]]. In fact, Mao et al., 2013 [71] illustrated the “H”-shaped morphology in *C. sinensis* hyphae that genetically contained either AT-biased Genotype #4 or #5 of *O. sinensis* without cooccurrence of the GC-biased Genotype #1 *H. sinensis* and that the genome-independent AT-biased *O. sinensis* genotypes shared indistinguishable *H. sinensis*-like morphological and growth characteristics. To date, no study has reported the identification of a- and α-pheromone genes in the genome or transcriptome assemblies of *H. sinensis* strains or in the metatranscriptome assemblies of natural *C. sinensis*; however, α- and a- pheromone receptor genes were found to differentially occur in the genome and transcriptome assemblies of *H. sinensis* and in metatranscriptome assemblies of the natural *C. sinensis* insect–fungi complex [Hu et al., 2013 [31]; Li et al., 2024 [37]]. Thus, the mating signal transduction between sexual partners to ensure appropriate interaction between MAT1-1-1 and MAT1-2-1 proteins with functional stereostructures remains a mystery in *O. sinensis*, which may inspire further research.

### 4.5. Heteromorphic 3D Structures of the Mating Proteins and Sexual Reproduction Strategy During the Lifecycle of the Natural C. sinensis Insect–Fungi Complex

In addition to the intensive attention given to *H. sinensis*, the N-terminally and midsequence-truncated MAT1-1-1 proteins and the variable MAT1-2-1 proteins encoded by the metatranscriptome assemblies of the natural *C. sinensis* insect–fungi complex exhibit alterations in the 2D structures of the proteins (*cf*. Figure 7, Figure 8, Figure 9 and Figure 10). These results suggest heteromorphic 3D structures of the mating proteins and dysfunctional or anomalous fungal mating processes during the lifecycle of the natural *C. sinensis* insect–fungi complex.

Li et al., 2016 [13] reported the genetic heterogeneity of the wild-type *C. sinensis* isolates CH1 and CH2, which were isolated from the intestines of healthy living larvae of *Hepialus lagii* Yan, based on their *H. sinensis*-like morphology and growth characteristics. The *C. sinensis* isolates CH1 and CH2 contained GC-biased Genotype #1 (*H. sinensis*) and AT-biased Genotypes #4–5 of *O. sinensis*, as well as *Paecilomyces hepiali* [Dai et al., 1989 [81]], which was renamed *Samsoniella hepiali* in 2020 [Wang et al., 2020 [82]]. The impure wild-type *C. sinensis* isolates exhibited 15–39-fold greater inoculation potency on the larvae of *Hepialus armoricanus* than did pure *H. sinensis* (n = 100 for each inoculant; *p* < 0.001), indicating the symbiosis of multiple intrinsic fungi during the lifecycle of natural *C. sinensis*, at least in the larva infection stage [Li et al., 2016 [13]].

We found that no repetitive copies of the *MAT1-1-1* or *MAT1-2-1* genes were identified in the *H. sinensis* genome assemblies. Thus, the genetic heterogeneity of the wild-type *C. sinensis* isolates suggests that the coexisting MAT1-1-1 and MAT1-2-1 proteins with varied amino acid sequences and diverse 3D structures detected in 24.3% of the wild-type *C. sinensis* isolates might have been derived from genome-independent heterospecific fungi, which may pair complementarily and reciprocally to ensure proper interactions and accomplish mating processes during the heterothallic or hybrid reproduction of *O. sinensis*. The different fungal sources for the cooccurring MAT1-1-1 and MAT1-2-1 proteins are evidenced by the species contradiction between the inoculants (3 purified GC-biased *H. sinensis* strains) and the genome-independent teleomorphic AT-biased Genotype #4, which is reportedly the sole teleomorphic fungus present in the fruiting body of cultivated *C. sinensis* [Wei et al., 2016 [19]]. These authors reported that successful artificial inoculation-based cultivation projects under product-oriented industrial settings used a special cultivation strategy without pursuing strict fungal purification by adding soil that was transported from natural *C. sinensis* production areas on the Qinghai–Tibet Plateau into the industrial cultivation system [Wei et al., 2016 [19]].

The AMT1-1-1 and MAT1-2-1 proteins with diverse stereostructures may originate from heterogeneous fungal sources in single hyphal and ascosporic cells. Li et al., 2013 [70] reported the genetic heterogeneity of 15 cultures of two groups derived from monoascospores, the reproductive cells of natural *C. sinensis*, after 25 days of in vitro incubation at 18 °C. The first group included seven homogeneous clones (1207, 1218, 1219, 1221, 1222, 1225, and 1229), containing only *H. sinensis* (GC-biased Genotype #1 of *O. sinensis*). The second group included eight heterogeneous clones (1206, 1208, 1209, 1214, 1220, 1224, 1227, and 1228), containing both GC-biased Genotype #1 and AT-biased Genotype #5 of *O. sinensis*. The sequences of the GC- and AT-biased *O. sinensis* genotypes reside in independent genomes and belong to independent fungi [Xiao et al., 2009 [83]; Zhu et al., 2010 [84]; Li et al., 2022 [6], 2024 [37]]. Bushley et al., 2013 [13] The collaborators of Li et al., 2013 [70], observed multicellular heterokaryotic hyphae and ascospores of natural *C. sinensis* with mononucleated, binucleated, trinucleated, and tetranucleated structures and reported the PCR results for 22 clones, which included 7 additional clones (1210, 1211, 1212, 1213, 1216, 1223, and 1226) forming the third group of ascosporic clones in addition to the aforementioned first and second groups of clones (*cf*. Figure 2 and Figure 3 of [Bushley et al., 2013 [32]]). However, neither Bushley et al., 2013 [32] nor Li et al., 2013 [70] reported the genetic features of the third group of ascosporic clones. However, there is no doubt from the two literature reports that the ascosporic clones from the third group were genetically distinct from those in the homogeneous first group or heterogeneous second group and apparently belonged to taxonomically different fungal species or fungal complexes.

Zhang and Zhang [2015 [35]] commented that the nuclei of binucleated hyphal and ascosporic cells (as well as mononucleated, trinucleated, and tetranucleated cells) of natural *C. sinensis* likely contained different genetic materials. These multicellular hyphal and ascosporic cells of natural *C. sinensis* might contain two or more sets of genomes of independent fungi, which might be responsible for the production of complementary mating proteins for sexual reproductive outcrossing. Thus, the monoascospores of natural *C. sinensis* might be characterized by more complex genetic heterogeneity, coexisting with more heterospecific fungal species than the cooccurring GC-biased Genotype #1 and AT-biased Genotype #5, which were reported by Li et al., 2013 [70].

Unlike culture-dependent experiments, which are apparently unable to detect nonculturable fungal species, Li et al., 2023 [49], 2023 [15] reported culture-independent studies and demonstrated the genetic heterogeneity of *C. sinensis* ascospores and the stromal fertile portion (SFP), which is densely covered with numerous ascocarps, which are the reproductive cells and organs of natural *C. sinensis*. These authors observed semi-ejected and fully ejected ascospores of natural *C. sinensis* and reported the cooccurrence of the GC-biased *O. sinensis* Genotypes #1 and #13/14, AT-biased *O. sinensis* Genotypes #5–6 and #16 within AT-biased clade A in the Bayesian phylogenetic tree, *S. hepiali* (≡*P. hepiali*), and an AB067719-type fungus. In addition, the *C. sinensis* SFP contained another fungal group, namely, AT-biased Genotypes #4 and #15 of *O. sinensis*, which were clustered into AT-biased clade B in the Bayesian phylogenetic tree [Li et al., 2023 [15]]. Genotypes #4 and #15 were absent in the ascospores, which is consistent with the results of culture-dependent studies [Li et al., 2013 [70]]. The abundance of fungal components exhibited marked dynamic alterations in a disproportional and asynchronous manner in the *C. sinensis* SFP before and after ascospore ejection and in the two types of ascospores [Li et al., 2023 [49]]. Thus, the coexistence of the MAT1-1-1 and MAT1-2-1 proteins detected from the wild-type *C. sinensis* isolates might have been derived interindividually from fungi that might serve as mating partners of *O. sinensis* to accomplish heterothallic or hybrid reproduction.

Li et al., 2023 [36], 2024 [37] summarized prior scientific evidence regarding the sexual reproduction of *O. sinensis*. Based on genetic heterogeneity with multiple heterospecific fungal species in the natural *C. sinensis* insect–fungi complexes and multicellular heterokaryotic structures of ascospores and hyphae, the scientific evidence may be divided into two aspects: (1) the 17 cooccurring genome-independent genotypes of *O. sinensis* in different combinations and (2) the taxonomically heterospecific fungal species, which are based on the mycobiota of >90 co-colonizing fungi belonging to at least 37 fungal genera in the stromata and caterpillar bodies of natural *C. sinensis* insect–fungi complexes [Zhang et al., 2010 [9], 2018 [10]; Xiang et al., 2014 [45]; Meng et al., 2015 [85]; Xia et al., 2015 [11], 2017 [46]; Guo et al., 2017 [12]; Wang et al., 2018 [86]; Zhong et al., 2018 [16]; Li et al., 2019 [87]; Zhao et al., 2020 [88]; Yang et al., 2021 [89]; Kang et al., 2024 [17]].

(1)Evidence for the differential cooccurrence of multiple genotypes of *O. sinensis* in the compartments of the natural *C. sinensis* insect–fungi complex is as follows:(1-a).Differential occurrence of AT-biased Genotype #4 or #5 of *O. sinensis* without the cooccurrence of GC-biased *H. sinensis* in natural *C. sinensis* samples collected from different production areas in geographically remote locations [Engh 1999 [90]; Kinjo and Zang 2001 [68]; Stensrud et al., 2005 [91], 2007 [92]; Mao et al., 2013 [71]];(1-b).Multiple cooccurring GC- and AT-biased genotypes of *O. sinensis* have been observed differentially in different combinations in the stroma, caterpillar body, ascocarps, and ascospores of natural *C. sinensis* [Xiao et al., 2009 [83]; Zhu et al., 2010 [84]; Li et al., 2013 [70], 2022 [6], 2023c [49], 2023d [15]; Mao et al., 2013 [71]]. The abundances of the *O. sinensis* genotypes underwent dynamic alterations in an asynchronous, disproportional manner in the caterpillar bodies and stromata of *C. sinensis* specimens during maturation, with a consistent predominance of the AT-biased genotypes of *O. sinensis*, not the GC-biased *H. sinensis*, in the stromata, indicating that the sequences of *O. sinensis* genotypes were present in independent genomes of different fungi [Xiao et al., 2009 [83]; Zhu et al., 2010 [84]; Hu et al., 2013 [31]; Li et al., 2013 [70], 2016a [40], 2020 [14], 2022 [6], 2023c [49]; Jin et al., 2020 [41]; Liu et al., 2020 [42]; Shu et al., 2020 [43]];(1-c).The GC-biased Genotypes #1 and #2 of *O. sinensis* cooccur in the stromata of natural *C. sinensis*. The abundance of the GC-biased genotypes was dynamically altered during *C. sinensis* maturation [Zhu et al., 2010 [84]];(1-d).The cooccurrence of GC-biased genomically independent Genotypes #1 and #7 of *O. sinensis* was detected in the same specimen of natural *C. sinensis* [Chen et al., 2011 [69]];(1-e).A species contradiction between the anamorphic inoculants (GC-biased Genotype #1 *H. sinensis* strains) and the sole teleomorph of AT-biased Genotype #4 of *O. sinensis* was detected in the fruiting body of cultivated *C. sinensis* in a product-oriented industrial setting [Wei et al., 2016 [19]];(1-f).Discovery of Genotypes #13 and #14 of *O. sinensis* in semi-ejected and fully ejected multicellular heterokaryotic ascospores, respectively, collected from the same *C. sinensis* samples [Li et al., 2023 [15]];(1-g).The genetic heterogeneity of ascospores and SFP, the reproductive cells and organs of natural *C. sinensis*, involves multiple GC- and AT-biased *O. sinensis* genotypes in different combinations [Li et al., 2013 [70], 2022 [6], 2023 [49], 2023 [15]].

(2)Evidence for the differential cooccurrence of heterospecific fungal species in different compartments of the natural *C. sinensis* insect–fungi complex is as follows:(2-a).Mycobiota findings for differential cooccurrence of >90 fungal species of at least 37 fungal genera in the caterpillar bodies and stromata of natural *C. sinensis* [Zhang et al., 2010 [9], 2018 [10]; Xia et al., 2015 [11]; Guo et al., 2017 [12]; Zhong et al., 2018 [16]; Kang et al., 2024 [17]];(2-b).A good number of *C. sinensis* isolates contained mutant MAT1-1-1 and MAT1-2-1 proteins, especially those proteins with C- and/or N-terminal truncations that belong to nine and four diverse 3D structural morphs (*cf*. Figure 4 and Figure 6), respectively. The mutant proteins were either clustered into a separate Bayesian clade or clustered within the main clustering branches in the Bayesian trees (*cf*. Figure 1 and Figure 2). The MAT1-1-1 and MAT1-2-1 proteins encoded by metatranscriptome assemblies of natural *C. sinensis* also exhibited either large-segment truncation or sequence variations similar to those observed in wild-type *C. sinensis* isolates (*cf*. Figure 7, Figure 8, Figure 9 and Figure 10). Some of the mutant proteins might be produced by heterospecific fungi in impure wild-type *C. sinensis* isolates and in natural *C. sinensis* insect–fungi complexes;(2-c).Discoveries of the formation of the heterospecific *Cordyceps*-*Tolypocladium* complex in natural *C. sinensis* [Engh 1999 [90]; Stensrud et al., 2005 [91], 2007 [92]] and the dual anamorphs of *O. sinensis*, involving psychrophilic *H. sinensis* and mesophilic *Tolypocladium sinensis* [Li 1988 [93]; Chen et al., 2004 [94]; Leung et al., 2006 [95]; Barseghyan et al., 2011 [96]];(2-d).A close association of psychrophilic *H. sinensis* and mesophilic *S. hepiali* (≡*P. hepiali*) was found in the caterpillar body, stroma, ascospores, and stromal fertile portion, which was densely covered with numerous ascocarps of natural *C. sinensis*, and in the wild-type *C. sinensis* complexes, which appeared to be difficult to purify [Dai et al., 1989 [81]; Jiang & Yao 2003 [8]; Chen et al., 2004 [94]; Zhu et al., 2007 [97], 2010 [84]; Yang et al., 2008 [98]; Li et al., 2016 [13], 2023 [15]];(2-e).Although Genotypes #13–14 are among the 17 genotypes of *O. sinensis*, these 2 GC-biased genotypes feature precise reciprocal cross substitutions of large DNA segments among two heterospecific parental fungi, namely, *H. sinensis* and an AB067719-type fungus. The taxonomic position of the AB067719-type fungus is undetermined to date, and more than 900 heterospecific fungal sequences, which are highly homologous to AB067719, have been uploaded to GenBank [Li et al., 2023 [15]]. Chromosomal intertwining and genetic material recombination may occur after plasmogamy and karyogamy of the heterospecific parental fungi under sexual reproduction hybridization or parasexuality, which is characterized by the prevalence of heterokaryosis and results in concerted chromosome loss for transferring–substituting genetic materials without conventional meiosis [Bennett & Johnson 2003 [99]; Sherwood & Bennett 2009 [100]; Bushley et al., 2013 [32]; Seervai et al., 2013 [73]; Nakamura et al., 2019 [74]; Mishra et al., 2021 [78]; Kück et al., 2022 [80]; Li et al., 2023 [15]].

The selection of different genotypes of *O. sinensis* or heterospecific fungal species as sexual partners depends on their mating choices for hybridization or parasexuality and their ability to break interspecific isolation barriers to adapt to extremely harsh ecological environments on the Qinghai–Tibet Plateau and the seasonal climate changes from extremely cold winters, when *C. sinensis* is in its asexual growth phase, to warmer springs and early summers, when *C. sinensis* switches to the sexual reproduction phase [Pfennig 2007 [72]; Du et al., 2020 [75]; Hėnault et al., 2020 [76]; Samarasinghe et al., 2020 [77]; Steensels & Gallone 2021 [79]].

## 5. Conclusions

The analysis of the MAT1-1-1 and MAT1-2-1 proteins in the 173 *H. sinensis* strains and wild-type *C. sinensis* isolates revealed heteromorphic stereostructures of the mating proteins, which were clustered into multiple Bayesian clustering clades and branches. In addition to the evidence of alternative splicing and differential occurrence and translation of the *MAT1-1-1* and *MAT1-2-1* genes in *H. sinensis* [Li et al., 2023 [36], 2024 [37]], the diversity of heteromorphic mating proteins suggested stereostructure-related alterations in the mating functions of proteins and provided additional evidence supporting the self-sterility hypothesis under heterothallic and hybrid reproduction for *O. sinensis*, including *H. sinensis*, Genotype #1 of the 17 genome-independent *O. sinensis* genotypes. The heteromorphic stereostructures of the mutant MAT1-1-1 and MAT1-2-1 proteins discovered in wild-type *C. sinensis* isolates and the natural *C. sinensis* insect–fungi complexes may suggest diverse sources of the mating proteins produced by two or more cooccurring heterospecific fungal species in natural *C. sinensis* that have been discovered in mycobiota, molecular, metagenomic, and metatranscriptomic studies, regardless of whether culture-dependent or culture-independent research strategies were used.

## Figures and Tables

**Figure 1 jof-11-00244-f001:**
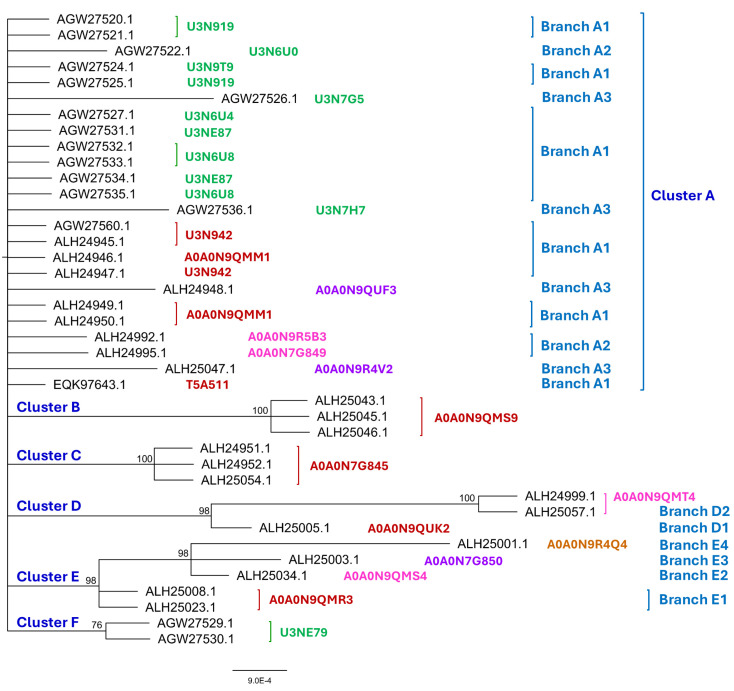
The Bayesian majority rule consensus clustering tree was inferred via MrBayes v3.2.7 software for the 40 full-length and truncated MAT1-1-1 proteins of the *H. sinensis* strains and wild-type *C. sinensis* isolates. The clusters and their branches (in blue) are shown alongside the tree. The AlphaFold UniProt codes for the 3D structures of the full-length proteins are shown in red alongside the tree for Branch 1 of the clusters, in pink for Branch 2, in purple for Branch 3, and in brown for Branch 4. The AlphaFold UniProt codes in green indicate the N-/C-terminally truncated MAT1-1-1 proteins.

**Figure 2 jof-11-00244-f002:**
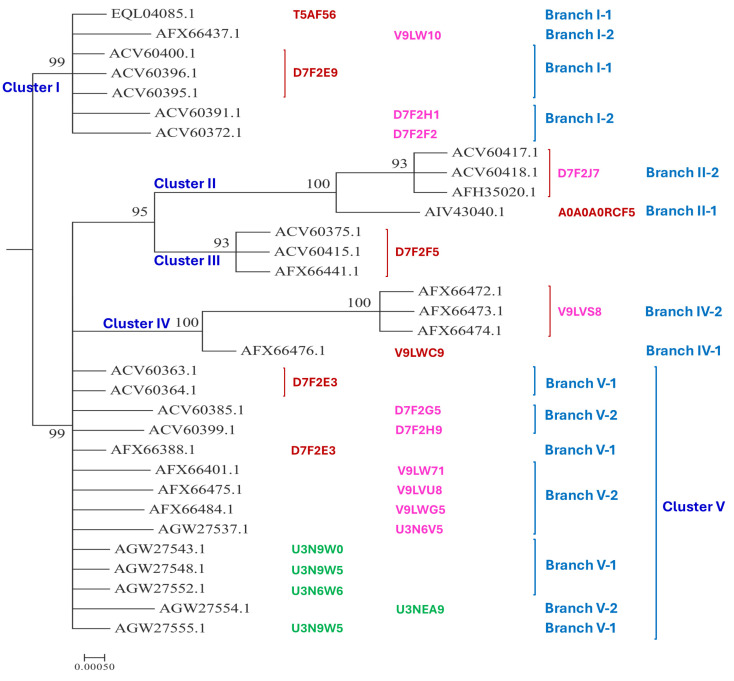
The Bayesian majority rule consensus clustering tree was inferred via MrBayes v3.2.7 software for the 32 full-length and truncated MAT1-2-1 proteins of the *H. sinensis* strains and wild-type *C. sinensis* isolates. The clusters and their branches (in blue) are shown alongside the tree. The AlphaFold UniProt codes for the 3D structures of the full-length proteins are shown in red alongside the tree for Branch 1 of the clusters and in pink for Branch 2 of the clusters. The AlphaFold UniProt codes in green indicate the C-terminally truncated MAT1-2-1 proteins.

**Figure 3 jof-11-00244-f003:**
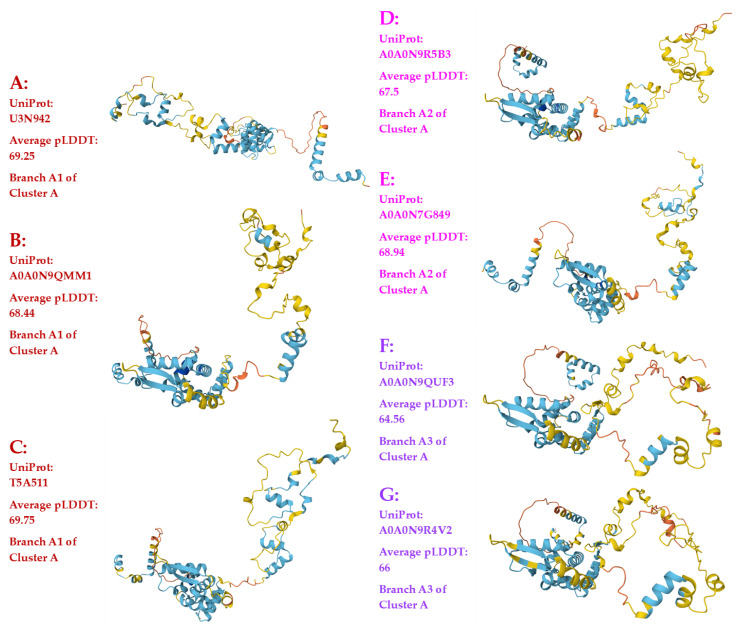
Fifteen 3D structural morphs (Panels **A**–**O**) for the 118 full-length MAT1-1-1 proteins of *H. sinensis* strains and wild-type *C. sinensis* isolates. The UniProt codes in red are for Branch 1 of each cluster shown alongside the Bayesian tree in Figure 1, while those in pink are for Branch 2, those in purple are for Branch 3, and those in brown are for Branch 4. An average pLDDT score for each predicted 3D structural model was computed via AlphaFold technology and is shown in each of the 3D structural panels (**A**–**O**), indicating model confidence (*cf*. Section 2.4): 
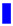
 very high (pLDDT > 90), 
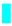
 high (90 > pLDDT > 70), 
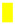
 low (70 > pLDDT > 50), and 
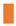
 very low (pLDDT < 50).

**Figure 4 jof-11-00244-f004:**
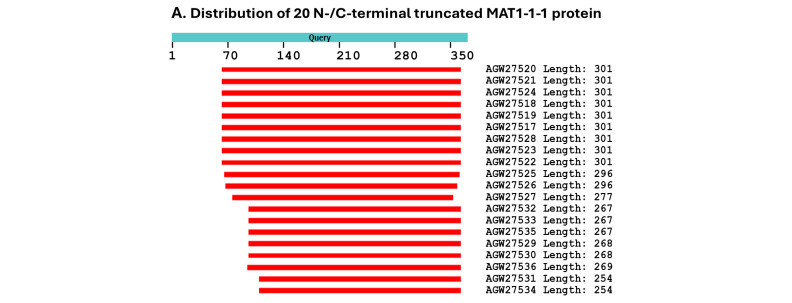
The sequence distribution (Panel **A**) and 9 diverse 3D structure morphs (Panels **B**–**J**) of the 20 N-/C-terminally truncated MAT1-1-1 proteins of *H. sinensis* strains and wild-type *C. sinensis* isolates. Model confidence: 
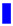
 very high (pLDDT > 90), 
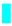
 high (90 > pLDDT > 70), 
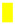
 low (70 > pLDDT > 50), and 
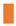
 very low (pLDDT < 50).

**Figure 5 jof-11-00244-f005:**
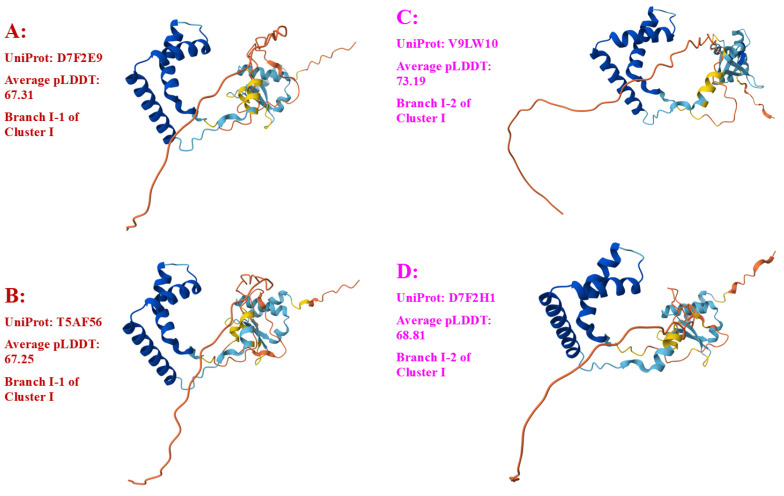
Seventeen 3D structural morphs (Panels **A**–**Q**) for the full-length MAT1-2-1 proteins of *H. sinensis* strains and wild-type *C. sinensis* isolates. The UniProt codes in red are for Branch 1 of each cluster shown in the Bayesian tree in Figure 2, and those in pink are for Branch 2. Model confidence: 
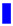
 very high (pLDDT > 90), 
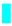
 high (90 > pLDDT > 70), 
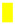
 low (70 > pLDDT > 50), and 
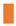
 very low (pLDDT < 50).

**Figure 6 jof-11-00244-f006:**
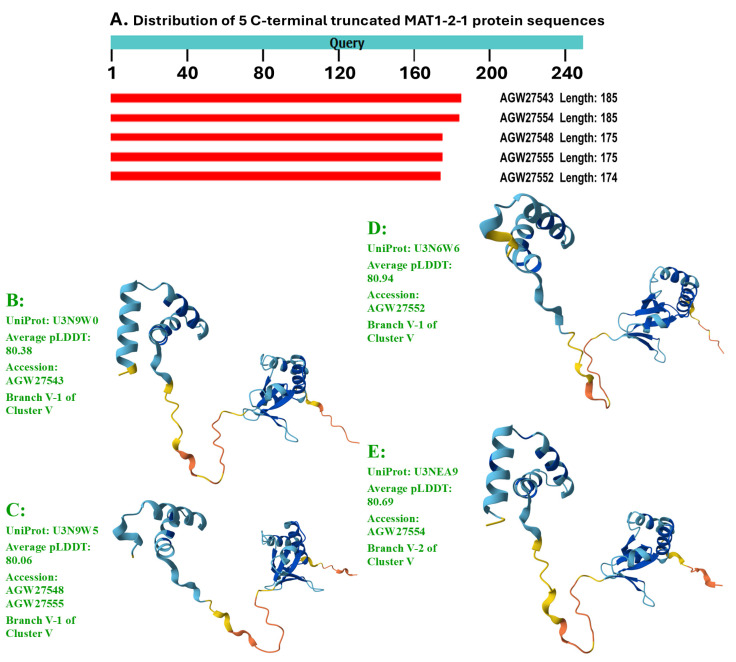
The sequence distribution (Panel **A**) and 3D structures (Panels **B–E**) of 5 C-terminally truncated MAT1-2-1 proteins of *H. sinensis* strains and wild-type *C. sinensis* isolates belonging to 4 diverse 3D structural morphs. Model confidence: 
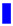
 very high (pLDDT > 90), 
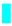
 high (90 > pLDDT > 70), 
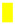
 low (70 > pLDDT > 50), and 
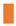
 very low (pLDDT < 50).

**Figure 7 jof-11-00244-f007:**
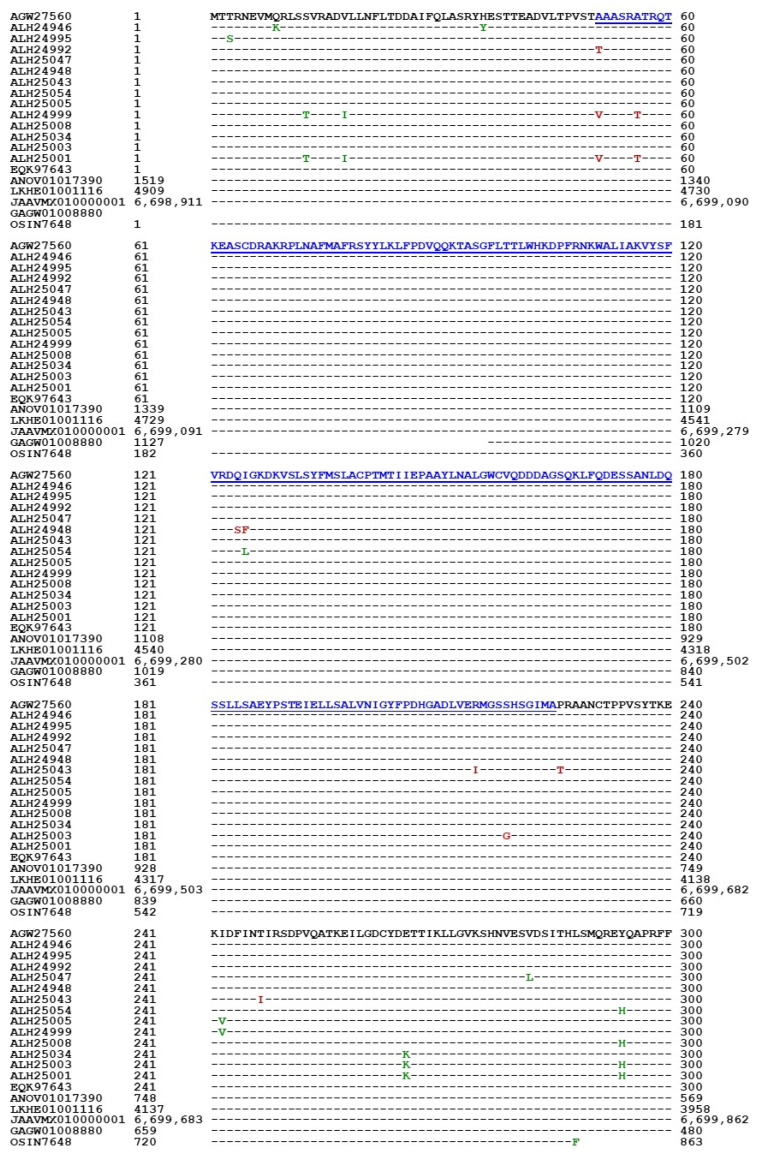
Alignment of the full-length sequences of representative MAT1-1-1 proteins of 15 structural morphs and corresponding translated sequence segments of the genome and metatranscriptome assemblies of *H. sinensis* strains and natural *C. sinensis*. The MATalpha_HMGbox domain is highlighted in blue and underlined in the query protein sequence AGW27560 (51→225). The residues shown in green indicate conservative amino acid substitutions, and those in red indicate nonconservative amino acid substitutions. The hyphens indicate identical amino acid residues, and the spaces denote unmatched protein sequence gaps.

**Figure 8 jof-11-00244-f008:**
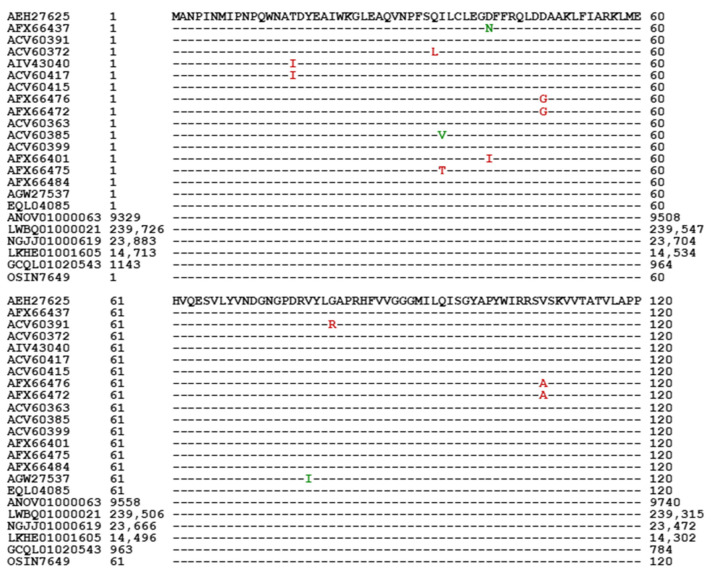
Alignment of the full-length sequences of representative MAT1-2-1 proteins of 17 diverse 3D structural morphs and the translated segments of the corresponding genome, transcriptome, and metatranscriptome assemblies of *H. sinensis* strains and natural *C. sinensis*. An HMG-box_ROX1-like domain is highlighted in blue and underlined in the query protein sequence AEH27625 (127→197). The residues shown in green refer to conservative amino acid substitutions, and those in red indicate nonconservative amino acid substitutions. The hyphens indicate identical amino acid residues.

**Figure 9 jof-11-00244-f009:**
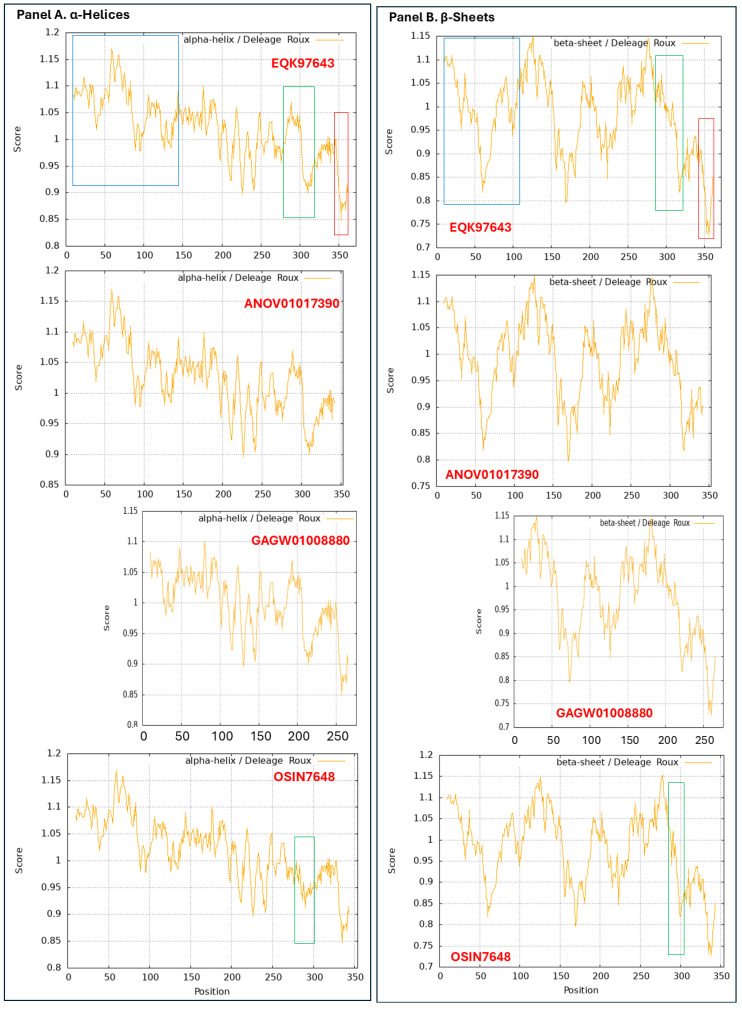
ExPASy ProtScale plots for the α-helices (Panel **A**), β-sheets (Panel **B**), β-turns (Panel **C**), and coils (Panel **D**) of the MAT1-1-1 proteins. Each panel contains 4 ProtScale plots for the 4 MAT1-1-1 proteins. The open boxes in blue in all EQK97643 plots indicate the N-terminal truncation region occurring in the MAT1-1-1 protein encoded by the metatranscriptome assembly GAGW01008880. The open boxes in red in all the EQK97643 plots indicate the C-terminal truncation region occurring in the genome assembly ANOV01017390. The open boxes in green in all the OSIN7648 plots, as well as in the corresponding region in all plots for the MAT1-1-1 protein EQK97643 for topology and waveform comparisons, indicate the midsequence truncation region occurring in the MAT1-1-1 protein encoded by the metatranscriptome assembly OSIN7648.

**Figure 10 jof-11-00244-f010:**
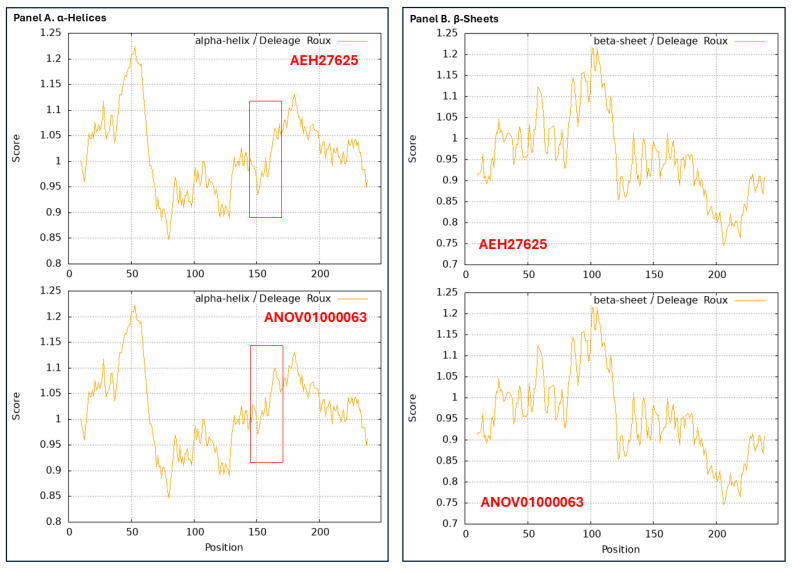
ExPASy ProtScale plots for the α-helices (Panel **A**), β-sheets (Panel **B**), β-turns (Panel **C**), and coils (Panel **D**) of the MAT1-2-1 proteins. Each panel contains 2 ProtScale plots. The open boxes in red in the ANOV01000063 plots in Panels (**A**,**C**,**D**) for the α-helices, β-turns, and coils, as well as in the corresponding region in the AEH27625 plots for the authentic MAT1-2-1 protein for topology and waveform comparisons, indicate the variation region occurring in the genome assembly.

**Table 1 jof-11-00244-t001:** GenBank accession numbers (in green, in parentheses) for the full-length MAT1-1-1 proteins of the *H. sinensis* strains and wild-type *C. sinensis* isolates under the corresponding AlphaFold UniProt codes.

AlphaFold UniProt Code (Bayesian Cluster/Branch *)	Strain/Isolate Number (GenBank Accession Number)
U3N942 (**A1**)	**CS68-2-1229** (AGW27560) (AGW27528), GS09_111 (ALH24945), GS09_131 (ALH24947), ID10_1 (ALH24954), IOZ07 (KAF4512729), NP10_1 (ALH24955), NP10_2 (ALH24956), QH07_188 (ALH24957), QH07_197 (ALH24958), QH09_37 (ALH24968), QH09_46 (ALH24969), QH09_56 (ALH24970), QH09_66 (ALH24971), QH09_78 (ALH24972), QH09_93 (ALH24973), QH09_122 (ALH24959), QH09_131 (ALH24960), QH09_151 (ALH24961), QH09_20L (ALH24965), QH09_33L (ALH24967), QH10_1 (ALH24974), QH10_4 (ALH24975), QH10_7 (ALH24976), SC09_21 (ALH24987), SC09_36 (ALH24988), SC09_37 (ALH24989), SC09_47 (ALH24990), SC09_57 (ALH24991), SC09_77 (ALH24993), SC09_107 (ALH24978), SC09_117 (ALH24979), SC09_128 (ALH24980), SC09_147 (ALH24981), SC09_157 (ALH24982), SC09_167 (ALH24983), SC09_180 (ALH24984), SC09_190 (ALH24985), SC09_200 (ALH24986), SC10_18 (ALH24996), SC10_21 (ALH24997), SC10_4 (ALH24998), XZ05_3 (ALH25002), XZ05_7 (ALH25004), XZ05_12 (ALH25000), XZ06_124 (ALH25006), XZ06_152 (ALH25007), XZ07_108 (ALH25009), XZ07_133 (ALH25010), XZ07_154 (ALH25011), XZ07_166 (ALH25012), XZ07_176 (ALH25013), XZ07_180 (ALH25014), XZ08_4 (ALH25018), XZ08_10 (ALH25015), XZ08_24 (ALH25016), XZ08_26 (ALH25017), XZ08_56 (ALH25019), XZ08_59 (ALH25020), XZ08_A1 (ALH25021), XZ08_B1 (ALH25022), XZ09_4 (ALH25029), XZ09_46 (ALH25030), XZ09_48 (ALH25031), XZ09_59 (ALH25032), XZ09_71 (ALH25033), XZ09_80 (ALH25055), XZ09_106 (ALH25024), XZ09_113 (ALH25025), XZ09_118 (ALH25026), XZ09_15 (ALH25027), XZ09_32 (ALH25028), XZ10_7 (ALH25038), XZ10_15 (ALH25035), XZ10_17 (ALH25036), XZ10_23 (ALH25037), XZ12_1 (ALH25056), XZ12_33 (ALH25058), XZ12_43 (ALH25059), YN07_6 (ALH25039), YN07_8 (ALH25040), YN09_3 (ALH25044), YN09_72 (ALH25049), YN09_81 (ALH25050), YN09_85 (ALH25051), YN09_89 (ALH25052), YN09_96 (ALH25053),YN09_101 (ALH25041), YN09_140 (ALH25042)
A0A0N9QMM1 (**A1**)	GS09_121 (ALH24946), GS09_201 (ALH24949), GS09_225 (ALH24950), SC09_1 (ALH24977)
T5A511 (**A1**)	**Co18** (EQK97643) (KE657544 410←1519 and ANOV01017390 410←1519)
A0A0N9R5B3 (**A2**)	SC09_65 (ALH24992)
A0A0N7G849 (**A2**)	SC09_97 (ALH24995)
A0A0N9QUF3 (**A3**)	GS09_143 (ALH24948)
A0A0N9R4V2 (**A3**)	YN09_61 (ALH25047)
A0A0N9QMS9 (**B**)	YN09_6 (ALH25046), YN09_22 (ALH25043), YN09_51 (ALH25045), YN09_64 (ALH25048)
A0A0N7G845 (**C**)	GS09_229 (ALH24951), GS09_281 (ALH24952), GS09_311 (ALH25054), GS10_1 (ALH24953), QH09_164 (ALH24962), QH09_173 (ALH24963), QH09_201 (ALH24964), QH09_210 (ALH24966), SC09_87 (ALH24994)
A0A0N9QUK2 (**D1**)	XZ05_8 (ALH25005)
A0A0N9QMT4 (**D2**)	XZ07_H2 (ALH24999), XZ12_16 (ALH25057)
A0A0N9QMR3 (**E1**)	XZ06_260 (ALH25008), XZ09_100 (ALH25023)
A0A0N9QMS4 (**E2**)	XZ09_95 (ALH25034)
A0A0N7G850 (**E3**)	XZ05_6 (ALH25003)
A0A0N9R4Q4 (**E4**)	XZ05_2 (ALH25001)

Note: The names of the pure *H. sinensis* strains are highlighted in bold and underlined, whereas those of the wild-type *C. sinensis* isolates are not. * Branch 1 is shown in red, Branch 2 in pink, Branch 3 in purple, and Branch 4 in brown, with the cluster codes (English letters) in parentheses determined via the Bayesian analysis (*cf*. Figure 1 below). The “←” arrows indicate sequences in the antisense strands of the genome of the *H. sinensis* strain Co18.

**Table 2 jof-11-00244-t002:** GenBank accession numbers (in green and in parentheses) for the full-length MAT1-2-1 proteins of the *H. sinensis* strains or wild-type *C. sinensis* isolates under the corresponding AlphaFold UniProt codes.

AlphaFold UniProt Code (Bayesian Cluster/Branch **)	Strain/Isolate Number (GenBank Accession Number)
D7F2E9 (**I-1**)	**CS2** (AEH27625) (ACV60400), **CS26-277** (AGW27541), **CS36-1294** (AGW27538), **CS37-295** (AGW27539), SC-2 (ACV60395), SC-4 (ACV60396), SC-5 (ACV60398), SC-7 (ACV60397), SC09-37 (AFH35019), SC09_47 (AFX66423), SC09_57 (AFX66424), SC09_77 (AFX66426), SC09_97 (AFX66428), XZ05_7 (AFX66442), XZ05_12 (AFX66444), XZ06_152 (AFX66445),XZ07_11 (AFX66447), XZ07_46 (AFX66448), XZ09_106 (AFX66464), XZ09_15 (AFX66455), YN09_101 (AFX66482), YN09_72 (AFX66477), XZ09_113 (AFX66465), XZ-LZ06-1 (ACV60369), XZ-LZ06-7 (ACV60370), XZ-LZ06-21 (ACV60371), XZ-LZ06-108 (ACV60373), XZ-LZ07-30 (ACV60377), XZ-LZ07-108 (ACV60379), XZ-ML-191 (ACV60376), YN-1 (ACV60390), YN-5 (ACV60392), YN-6 (ACV60393), YN-8 (ACV60394), YN09_81 (AFX66478), YN09_85 (AFX66479), YN09_89 (AFX66480)
T5AF56 (**I-1**)	**Co18** (EQL04085) (ANOV01000063 9329→10182)
V9LW10 (**I-2**)	SC09_200 (AFX66437)
D7F2H1 (**I-2**)	YN-4 (ACV60391)
D7F2F2 (**I-2**)	XZ-LZ06-61 (ACV60372)
A0A0A0RCF5 (**II-1**)	XZ12_16 (AIV43040)
D7F2J7 (**II-2**)	XZ05_8 (AFX66443), XZ06-124 (AFH35020), XZ-LZ07-H2 (ACV60418), XZ-LZ07-H1 (ACV60417)
D7F2F5 (**III**)	XZ05_2 (AFX66441), XZ06_260 (AFX66446), XZ09_80 (AFX66461), XZ09_95 (AFX66462), XZ09_100 (AFX66463), XZ-LZ05-6 (ACV60415), XZ-SN-44 (ACV60375),
V9LWC9 (**IV-1**)	YN09_64 (AFX66476)
V9LVS8 (**IV-2**)	YN09_6 (AFX66472), YN09_22 (AFX66473), YN09_51 (AFX66474)
D7F2E3 (**V-1**)	GS09_111 (AFX66388), CS560-961 (AGW275424), QH09-93 (AFH35018), XZ-NQ-154 (ACV60363), XZ-NQ-155 (ACV6036)
D7F2G5 (**V-2**)	QH-YS-199 (ACV60385)
D7F2H9 (**V-2**)	SC-3 (ACV60399)
V9LW71 (**V-2**)	QH09_11 (AFX66401)
V9LVU8 (**V-2**)	YN09_61 (AFX66475)
V9LWG5 (**V-2**)	ID10_1 (AFX66484)
U3N6V5 (**V-2**)	CS6-251 (AGW27537)
‡	NP10_1 (AFX66485), NP10_2 (AFX66486), YN09_3 (AFX66471), YN09_96 (AFX66481), YN09_140 (AFX66483)

Note: The names of the pure *H. sinensis* strains are highlighted in bold and underlined, whereas those of the wild-type *C. sinensis* isolates are not. ** Branch 1 is shown in red and Branch 2 in pink, with the cluster codes (Roman numerals) in the parentheses determined via the Bayesian analysis, as shown in Figure 2 below. ‡, The 5 MAT1-2-1 protein sequences are included in the GenBank database but not in the AlphaFold database (*cf*. Appendix A). The “→” arrow indicates the sequence in the sense strand of the genome of the *H. sinensis* strain Co18.

**Table 3 jof-11-00244-t003:** Summary of the full-length MAT1-1-1 protein sequence alignment results (mutant amino acids and the percent similarity vs. the likely authentic protein AGW27560), correlating with the Bayesian branches/clusters and the 3D structural models and associated with the AlphaFold UniProt codes.

Accession Number	% Similarity to AGW27560	Amino Acid Residue Substitution			Bayesian Cluster		3D Structure Model	AlphaFold UniProt Code
		Conservative	Nonconservative		Branch	Cluster		
AGW27560 ALH24946 EQK97643	100% 99.4% 100%	Q-to-K, H-to-Y			A1	A	A B C	U3N942 A0A0N9QMM1 T5A511
ALH24992 ALH24995	99.7%		A-to-T		A2	A	D E	A0A0N9R5B3 A0A0N7G849
		T-to-S						
ALH25047 ALH24948	99.4%		S-to-L		A3	A	F G	A0A0N9R4V2 A0A0N9QUF3
ALH25043	98.9%		R-to-I, P-to-T, T-to-I, G-to-A,			B	H	A0A0N9QMS9
ALH25054	99.4%	I-to-L	A-to-V			C	I	A0A0N7G845
ALH25005	99.2%	H-to-Y	P-to-H		D1	D	J	A0A0N9QUK2
ALH24999	98.1%	S-to-T, I-to-V, H-to-Y	A-to-V, A-to-T		D2	D	K	A0A0N9QMT4
ALH25008	99.7%	Y-to-H			E1	E	L	A0A0N9QMR3
ALH25034	99.4%	E-to-K, Y-to-H			E2	E	M	A0A0N9QMS4
ALH25003	99.2%	E-to-K, Y-to-H	S-to-G		E3	E	N	A0A0N7G850
ALH25001	98.4%	S-to-T, V-to-I, E-to-K, Y-to-H	A-to-V, A-to-T		E4	E	O	A0A0N9R4Q4

Note: The amino acid residue substitutions shown in red indicate nonconservative changes within the MATalpha_HMGbox domain of the MAT1-1-1 proteins. Other residue substitutions shown in black are located within or outside the MATalpha_HMGbox domain.

**Table 4 jof-11-00244-t004:** Summary of the full-length MAT1-2-1 protein sequence alignment results (mutant amino acids and percent similarity vs. the likely authentic protein AEH27625), correlating with the Bayesian branches/clusters and the 3D structural models and associated with the AlphaFold UniProt codes.

Accession Number	% Similarity to AEH27625	Amino Acid Residue Substitution			Bayesian Cluster		3D Structure Model	AlphaFold UniProt Code
		Conservative	Nonconservative		Branch	Cluster		
AEH27625 EQL04085	100% 100%				I-1	I	A B	D7F2E9 T5AF56
AFX66437 ACV60391 ACV60372	99.6%	D-to-N			I-2	I	C D E	V9LW10 D7F2H1 D7F2F2
			G-to-R Q-to-L					
AIV43040	97.6%	Y-to-H, M-to-I, Q-to-R, S-to-T	T-to-I, A-to-G		II-1	II	F	A0A0A0RCF5
ACV60417	97.6%	Y-to-H, M-to-I, S-to-T	T-to-I, T-to-A, A-to-G		II-2	II	G	D7F2J7
ACV60415	98.8%	Y-to-H, M-to-I	T-to-G			III	H	D7F2F5
AFX66476	98.8%	Y-to-H	D-to-G, V-to-A		IV-1	IV	I	V9LWC9
AFX66472	97.6%	Y-to-H, D-to-N, S-to-T	D-to-G, V-to-A, A-to-T		IV-2	IV	J	V9LVS8
ACV60363	99.6%	Y-to-H			V-1	V	K	D7F2E3
ACV60385	99.2%	I-to-V, Y-to-H			V-2	V	L	D7F2G5
ACV60399	99.2%	Y-to-H, Q-to-R			V-2	V	M	D7F2H9
AFX66401	99.2%	Y-to-H	D-to-I		V-2	V	N	V9LW71
AFX66475	99.2%	Y-to-H	I-to-T		V-2	V	O	V9LVU8
AFX66484	99.2%	Y-to-H	A-to-T		V-2	V	P	V9LWG5
AGW27537	99.2%	V-to-I, Y-to-H			V-2	V	Q	U3N6V5

Note: The amino acid residue substitutions shown in red indicate conservative changes within the HMG-box_ROX1-like domain of the MAT1-2-1 proteins. Other residue substitutions shown in black are located within or outside the HMG-box_ROX1-like domain.

**Table 5 jof-11-00244-t005:** Percentage similarity between the sequences EQK97643 and AEH27625 for the MAT1-1-1 and MAT1-2-1 proteins, respectively, and the mating protein sequences encoded by the genome assemblies of *H. sinensis* strains.

*H. sinensis* Strain	Genome Assembly Segment	Percentage Similarity	
		MAT1-1-1 (vs. EQK97643)	MAT1-2-1 (vs. AEH27625)
Co18	ANOV01017390 (410←1519)	99.7%	
	ANOV01000063 (9329→10,182)		99.6%
1229	LKHE01001116 (3799←4909)	99.7%	
	LKHE01001605 (13,860←14,713)		99.6%
IOZ07	JAAVMX010000001 (6,698,911→6,700,021)	99.7%	
	JAAVMX000000000		―
ZJB12195	LWBQ00000000	―	
	LWBQ01000021 (238,873←239,726)		99.6%
CC1406-20395	NGJJ00000000	―	
	NGJJ01000619 (23,030←23,883)		99.6%

Note: The “→” and “←” arrows indicate sequences in the sense and antisense strands of the genomes, respectively.

**Table 6 jof-11-00244-t006:** Percentage similarity between the sequences EQK97643 and AEH27625 for the MAT1-1-1 and MAT1-2-1 proteins, respectively, and the mating proteins encoded by the transcriptome assembly of *H. sinensis* strain L0106 and the metatranscriptome assemblies of natural *C. sinensis*.

*H. sinensis* Strain or Natural *C. sinensis*	Transcriptome or Metatranscriptome Assembly Segment	Percentage Similarity	
		MAT1-1-1 (vs. EQK97643)	MAT1-2-1 (vs. AEH27625)
*H. sinensis* strain L0106	GCQL00000000	―	
	GCQL01020543 (397←1143)		99.6%
Mature natural *C. sinensis* (Collected at Deqin, Yunnan)	OSIN7648 (1→1065)	94.9%	
	OSIN7649 (1→397)		100%
Natural *C. sinensis* * (Collected at Kangding, Sichuan)	GAGW01008880 (300←1127)	100%	
	GAGW00000000		―

Note: *, Natural *C. sinensis* samples of unknown maturation stage. The “→” and “←” arrows indicate sequences in the sense and antisense strands of the genomes, respectively.

## Data Availability

All sequence and 3D structure data are available in the GenBank and AlphaFold databases, except for one set of metatranscriptome sequences from natural *C. sinensis* that was uploaded to the repository database www.plantkingdomgdb.com/Ophiocordyceps_sinensis/data/cds/Ophiocordyceps_sinensis_CDS.fas (accessed from 18 May 2017 to 18 January 2018) by Xia et al., 2017 [58], which is currently inaccessible, but a previously downloaded cDNA file was used for mating protein analysis.

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
