# Peer review of "Three-Dimensional Structural Heteromorphs of Mating-Type Proteins in Hirsutella sinensis and the Natural Cordyceps sinensis Insect–Fungal Complex"

_jof, 2025, doi:10.3390/jof11040244_

Round 1
Reviewer 1 Report
The manuscript demonstrates significant originality by exploring the genetic heterogeneity and reproduction mechanisms of Ophiocordyceps sinensis (O. sinensis), emphasizing its association with multiple heterospecific fungal species. The content is highly significant as it addresses key questions regarding the reproduction, genetic diversity, and ecological adaptability of O. sinensis, a species of ecological and economic importance. The discovery of diverse MAT proteins, heterokaryotic structures, and co-occurring heterospecific fungi sheds light on its complex biology and potential reproductive strategies. However, I have some remarks and comments on this manuscript that need clarification before the work can be published.
Comments:
Line 129. “MrBayes v3.2.7” – For the tool this citation is missing: “Fredrik Ronquist, Maxim Teslenko, Paul van der Mark, Daniel L. Ayres, Aaron Darling, Sebastian Höhna, Bret Larget, Liang Liu, Marc A. Suchard, John P. Huelsenbeck, MrBayes 3.2: Efficient Bayesian Phylogenetic Inference and Model Choice Across a Large Model Space, Systematic Biology, Volume 61, Issue 3, May 2012, Pages 539–542 https://doi.org/10.1093/sysbio/sys029 “.
Lines 138–139. "downloaded from the AlphaFold database (accessed on 10/18/2024 to 12/31/2025) for structural polymorphism analysis in this study." – It seems you meant “accessed on 10/18/2024 to 12/31/2024”. Please verify this.
Lines 159-162. The text states that "138 MAT1-1-1 proteins and 79 MAT1-2-1 proteins" were derived from "173 H. sinensis strains and wild-type C. sinensis isolates." This implies that these numbers should total 173, which is not clearly explained.
Figure captions for Figures 3, 4, 5, and 6. Please clarify the meaning of “pLDDT” and explain how this parameter was evaluated. Include a reference to the relevant resource.
Line 247. The term “stereostructural morphs” needs clarification, as it is not a widely used term.
Lines 519-520. “for the heteromorphic structures of 79 MAT1-2-1 proteins (cf. Figures 2 and 4-5)” – However, according to the figure caption, Figure 4 shows 3D models for MAT1-1-1 proteins, not MAT1-2-1 proteins. Please verify. Similarly, at Line 523: “3D structural morph A of the MAT1-2-1 proteins (cf. Figure 4)” – this does not match the Figure 4 caption.
Lines 581-582. “which is endangered at Level 2” – It is unclear what “Level 2” refers to. Please provide an explanation for a broad audience and include a reference to the resource.
Line 621. “Seervai et al. 2013, Seervai et al. 2013” – This reference is repeated.
Line 651. “Samoneilla hepiali” – This is likely a typo and should read “Samsoniella hepiali.” Additionally, please ensure that all Latin species names throughout the text are italicized.
3.6. Primary sequence – this is not correct phrase, should be primary structure or amino acid sequence
Some of the Figures should be moved to supplementary material.
Please see above.
Author Response
Responses to Reviewer #1:
1). You commented, “Line 129. “MrBayes v3.2.7” – For the tool this citation is missing: “Fredrik Ronquist, Maxim Teslenko, Paul van der Mark, Daniel L. Ayres, Aaron Darling, Sebastian Höhna, Bret Larget, Liang Liu, Marc A. Suchard, John P. Huelsenbeck, MrBayes 3.2: Efficient Bayesian Phylogenetic Inference and Model Choice Across a Large Model Space, Systematic Biology, Volume 61, Issue 3, May 2012, Pages 539–542 https://doi.org/10.1093/sysbio/sys029.”
Thanks. We added this information (ref. #58) to the reference list (Lines 1154–1156).
2). You commented, “Lines 138–139. "downloaded from the AlphaFold database (accessed on 10/18/2024 to 12/31/2025) for structural polymorphism analysis in this study." – It seems you meant “accessed on 10/18/2024 to 12/31/2024”. Please verify this.”
Thanks. We made the correction (Line 127).
3). You commented, “Lines 159–162. The text states that "138 MAT1-1-1 proteins and 79 MAT1-2-1 proteins" were derived from "173 H. sinensis strains and wild-type C. sinensis isolates." This implies that these numbers should total 173, which is not clearly explained.”
We explained the situation of possible conflicts in the numbers:
Among the 173 strains/isolates, 42 (24.3%) had records of AlphaFold-predicted 3D structures for both the MAT1-1-1 and MAT1-2-1 proteins. A majority (75.7%) of the strains/isolates presented 3D structure records for either the MAT1-1-1 or MAT1-2-1 protein, suggesting differential cooccurrences of the 2 mating proteins essential for the sexual reproduction of O. sinensis. In addition, strains CS68-2-1229 and CS2 have duplicated accession numbers for either the MAT1-1-1 or MAT1-2-1 protein, unlike the 171 other strains/isolates (i.e., 173=138+79–42–2). (Lines 182–188)
4). You commented, “Figure captions for Figures 3, 4, 5, and 6. Please clarify the meaning of “pLDDT” and explain how this parameter was evaluated. Include a reference to the relevant resource.”
The description of the per-residue model confidence in the predicted Local Distance Difference Test (pLDDT) is provided in the Methods section:
The AlphaFold database provides per-residue model confidence, the prediction of its score in the local distance difference test (pLDDT), between 0 and 100, a per-residue score that is assigned to each individual residue [Mariani et al. 2013; Jumper et al. 2021; David et al. 2022; Monzon et al. 2022; Xu et al. 2023; Abramson et al. 2024; Varadi et al. 2024]. Model confidence bands are used to color-code the residues in the 3D structure: very high confidence (pLDDT > 90) residues are shown in dark blue, high (90 > pLDDT > 70) in light blue, low (70 > pLDDT > 50) in yellow, and very low (pLDDT < 50) in orange [Mariani et al. 2013; Wroblewski & Kmiecik 2024]. Note that a protein region that is assigned a low pLDDT score does not necessarily indicate that this region is the most variable region in the protein sequence; in contrast, a substantially variable region of a protein may be assigned a high pLDDT score. The AlphaFold database provides an average pLDDT score for each of the predicted 3D structure models of mating proteins, representing the overall model confidence in the predicted 3D structures. (Lines 146-158)
5). You commented, “Line 247. The term “stereostructural morphs” needs clarification, as it is not a widely used term.”
We revised this phrase. The phrase “remaining 9 stereostructural morphs” was revised to “the remaining 9 diverse morphs of 3D structures” (Lines 305, 336, 637, and 646).
6). You commented, “Lines 519–520. “for the heteromorphic structures of 79 MAT1-2-1 proteins (cf. Figures 2 and 4-5)” – However, according to the figure caption, Figure 4 shows 3D models for MAT1-1-1 proteins, not MAT1-2-1 proteins. Please verify. Similarly, at Line 523: “3D structural morph A of the MAT1-2-1 proteins (cf. Figure 4)” – this does not match the Figure 4 caption.”
We apologize for the typos. It should be (cf. Figures 2 and 5–6) and (cf. Figure 5), (Lines 645 and 649), rather than (cf. Figures 2 and 4–5) and (cf. Figure 4), respectively. We corrected this in the text.
7). You commented, “Lines 581–582. “which is endangered at Level 2” – It is unclear what “Level 2” refers to. Please provide an explanation for a broad audience and include a reference to the resource.”
The China Ministry of Agriculture and Rural Affairs published a list of “National Key Protected Wild Plants” in 1999 and revised the list in 2021 on the basis of the following criteria:
Species with very few plants, extremely small wild populations, narrow distribution ranges, and brinks of extinction.
Endangered or rare species with significant economic, scientific, or cultural value;
Wild populations of important crops and genetically valuable close relatives;
Species that have significant economic value but have experienced a sharp decline in wild resources owing to excessive development and utilization pose a threat or serious threat to their survival.
The natural Cordyceps sinensis insect-fungi complex has been listed as a Level II natural species.
We added a reference (Ref. #7) for the list of “National Key Protected Wild Plants” (Lines 1034-1036).
8). You commented, “Line 621. “Seervai et al. 2013, Seervai et al. 2013” – This reference is repeated.”
Yes. We deleted that repeated citation (Line 779).
9). You commented, “Line 651. “Samoneilla hepiali” – This is likely a typo and should read “Samsoniella hepiali.” Additionally, please ensure that all Latin species names throughout the text are italicized.”
Thanks. We made a correction (Line 813).
10). You commented, “Primary sequence – this is not correct phrase, should be primary structure or amino acid sequence.”
Thanks. We made corrections (Lines 343–345, 419).
11). You commented, “Some of the Figures should be moved to supplementary material.”
Yes. We moved Table S1 and Figure S1 to the supplementary material.

Reviewer 2 Report
The manuscript by Li et al. presents comparisons of the sequences and 3D structure models for many versions of the MAT proteins from many Hirsutella sinensis and Cordyceps sinensis isolates. The authors do a good job of introducing that this fungal-insect complex is very genetically heterogeneous, that there are historical and continuing issues with how to refer to/classify the complex and independent fungi, and there is debate about the sexual cycle for the complex. They then go on to provide a comprehensive comparison of sequences and 3D models for variants of the MAT1-1-1 and MAT1-2-1 proteins available in databases to illustrate the diversity of sequences and structures. This work is an extension of prior work from the group showing differential expression and presence of different splice or deletion variants of the MAT1-1-1 and MAT1-2-1 genes for C. sinensis. The current work involves clustering and multiple alignments of variant protein sequences for each gene from many fungal isolates, showing AlphaFold 3D models representing the diversity of protein variants for each gene, and analyzing secondary structures throughout some variant truncated proteins. The authors try to be comprehensive by including all relevant structural models and clustering all representative sequences, including truncated versions of the proteins. Overall, they suggest that the diversity of protein sequences and structures indicates that there may be many MAT proteins that are not functional in these fungi, which argues against homothallism and supports heterothallism and/or hybridization for sexual reproduction in H. sinensis and C. sinensis. While the analysis presented is potentially useful for readers interested in this particular fungal-insect complex and sexual reproduction in fungi, my main concerns are with the presentation and organization of the information. I expect that many readers would be overwhelmed by the large amount of information shown in many figures and some tables and the absence of the authors providing good summaries of those analyses and highlighting the most important points to consider for those analyses in the Results section. The Discussion is very long and seems to develop almost into a review of the literature on sexual reproduction in H. sinensis and C. sinensis, rather than a focused discussion on the analyses presented in the current manuscript. I think that there need to be some major changes to the organization and presentation of the analyses, and then a revised manuscript can be reconsidered for publication in JoF.
1. The current organization of information in the Results section could make it hard for readers to follow along and appreciate the overall analyses. Currently, the authors start with the clustering of protein sequences, showing the variety of 3D structure models, then show a multiple sequence alignment, summary of the presence and sequence similarity of each gene from several genome assemblies of H. sinensis strains, summary of presence and sequence similarity of each gene in transcriptome or metatranscriptome data for H. sinensis or C. sinensis, and finish with analyses of secondary structures in truncated versions of the proteins. It seems that it would be more logical to begin with information about the presence and sequence variations of the genes in the genomes and transcriptomes, then move to protein sequence comparisons, and then discuss 3D protein structures and the analysis of secondary structures. For instance, if a gene is not present in the genome assembly or transcriptome, then clearly there would be no expectation for the protein to be present in an isolate, so it seems it would make more sense to start with that information, rather than start with presence/absence and variations in 3D protein structures and later explain that not all isolates have both genes in their genomes.
2. While it is admirable that the authors are trying to be comprehensive, it will be hard for readers to process so many references to various accession codes without frequent reminders of the relevance of mentioning those codes. For instance, the authors discuss both H. sinensis and C. sinensis throughout the paper, and occasionally refer to particular strains/isolates for each, but a reader looking at Table 1 would not have any idea of which isolates are H. sinensis and which are C. sinensis. That limits readers from appreciating which protein variants are in H. sinensis versus other genotypes. In lines 167-174, the authors refer to a couple specific strains and accession codes/numbers for those strains, including when data were deposited, but it would be unclear to a reader why those two strains and those accession numbers are being discussed in detail. Are these the most representative strains and protein sequences, for instance? Or are these the protein sequences that are most likely to be functional? Late in the Results section, the authors refer to “authentic” versions of the proteins (lines 420 and 437) and then again refer to “authentic” version in the Discussion (line 515). If the authors have reason to expect that particular accession codes/numbers are likely to represent the functional versions of the proteins, then that should be mentioned very early during the presentation of the analysis of protein variants in the Results – both the reasoning for referring to them as authentic and distinguishing which variants are being presumed to be functional. Then, the authors should be highlighting in their discussion of their analyses in the Results what the features (sequences/structures) of the presumably functional versions of the proteins are compared to other variants that are presumed to be defective. Without helping readers in this way, readers may struggle to understand what sequence and structural features are potentially/likely important for function of the proteins.
3. There need to be more summaries of what analyses are showing in the Results section that would help readers appreciate what the authors have found through their work. For instance, lines 512-523 are good and concise summaries of the clustering analysis for each MAT protein, but those summaries should be in the Results, not in the Discussion. The Discussion would be a place to speculate on what is summarized, but readers need these types of summaries highlighting what analyses show in the Results in order to not become lost with all the information being presented. Lines 196-206 and 234-242 are pretty good for summarizing the clustering analyses, though adding more emphasis based on comment #2 about what authors are considering to be “authentic” protein variants would be helpful. In contrast, the authors present a large number of 3D structures in Figures 3-6, but no text in the Results compares any particular features of these structural morphs. For instance, are presumably authentic proteins displaying more of a particular secondary structure (have mostly helices rather than beta sheets, for instance?) or displaying particular patterns of secondary structures in certain regions of the protein? It is also important to acknowledge that these are predicted structures. The authors should address whether most of the variation in 3D structures is in regions of low confidence (low pLDDT) or if there is substantial variation even in regions of high model confidence. It would also be important to remind readers for MAT1-1-1 that 89 of the 118 full-length proteins have structure A. In Figure 3, panel A does not seem to have any more importance than any other panel, but nearly all the isolates with full-length proteins have that 3D structure, so it would be important to remind readers of that and devote at least a little Results text to discussing features of that potentially functional structure.
It would help to also discuss more of what is shown in the multiple sequence alignments and the ExPASy analysis. For instance, the variations in the multiple sequence alignments are not compared in any way to the 3D structures shown. Considering that there are relatively few differences (all protein sequences aligned in Figures 7 and 8 are >97% similar), there should be some attempt to address whether certain types of substitutions appear to contribute to certain types of changes in the 3D structures. That would help readers appreciate whether some of the conservative versus nonconservative changes are less or more likely to be producing 3D structural changes. Section 3.9 describes the sequences and accession codes that are being compared for secondary structures, including regions deleted in the truncated variants. However, the authors simply state that the analysis shows changes in the topology that might be important (lines 436-443). However, there is no mention of specific secondary structures that are most affected. Also, the AlphaFold model for the full-length protein is not compared to the 2D analysis of secondary structures to show how the ExPASy analysis compares to the AlphaFold model. For instance, it could help readers to highlight/describe which sequence regions of ALH24945 are predicted to be in what secondary structures from the AlphaFold model. That would help readers better appreciate the potential importance of some differences noted in the secondary structure analysis. Section 3.10 indicates that there were not many changes in the secondary structure analysis, so there is less of a need to highlight particular structures in that case.
4. The Discussion is very long, and much of it is not specifically focused on the new analyses presented in the manuscript. Sections 4.3 and 4.4 from lines 580-799 is mostly a review of other information about reproduction of H. sinensis and C. sinensis. In this large portion of the Discussion, only lines 639-644 and 759-769 are specifically focused on something from the results of the current manuscript. Also, a prior comment suggested moving some text from Section 4.1 to the Results section. Considering the main analyses shown and the title of the manuscript, it would be better if the Discussion was more focused on the structural variation being shown. The authors can still review some other information regarding sexual reproduction, but there should be more of a focus on the MAT proteins. For instance, the authors could speculate on the important structural features of the MAT proteins considering other related transcription factors. For instance, are there particular domains/structures that may contribute to DNA binding or to the proteins interacting with each other, based on comparisons to other similar proteins. It would also be good to relate the clustering and alignment of sequences to the variation of structures seen. For instance, is there correspondence between particular clusters in Figures 1 and 2 and particular variations in 3D structures? Perhaps there could be some speculation on the types of sequence variations that cause particular folds or secondary structures to change dramatically in the 3D models. Currently, the only main message that comes through from showing the 3D models is that there are many variant structures, but there is no discussion of what might be more important to pay attention to in the structures for future work. It would also be important to acknowledge that the wide variety of structures depends on the accuracy of the AlphaFold models, and not all of the predicted structures are of high confidence in Figures 3-6. Therefore, the models could be under or overrepresenting the diversity of the structures. Overall, I think that readers would benefit from a more detailed discussion of the protein variants to consider the structure/function relationship for the MAT proteins, which could form the basis for future work on these proteins and sexual reproduction for these fungi.
Round 2
Reviewer 1 Report
I have no comments.
I have no comments.
Reviewer 2 Report
The authors have addressed all my concerns well and have provided good arguments for any instances when they chose not to take my suggestions. With these changes and arguments to support the rationale for their presentation, I think that the paper is now suitable for publication in JoF.
I do not have any new detailed comments for the authors to address, as they have done a good job of addressing my initial concerns.